# CeLLM: Can Large Language Models Achieve the AI Virtual Cell?

## Abstract

High-throughput single-cell sequencing has enabled large-scale cellular profiling and spurred the development of single-cell foundation models. These models, typically pretrained on transcriptomic data, learn general-purpose cellular representations but remain limited in modality coverage, causal reasoning, and interpretability, thus falling short of the vision of an Artificial Intelligence Virtual Cell (AIVC). In parallel, large language models (LLMs) have demonstrated strong potential for unifying heterogeneous modalities, adapting to diverse tasks, and generating interpretable reasoning chains in natural language, making them promising candidates toward AIVC. Recent progress in applying LLMs to tasks such as cell annotation and perturbation prediction highlights this potential, yet key challenges persist, including insufficient task coverage, narrow evaluation metrics, and limited robustness to input and prompting factors. To address these gaps, we introduce **CeLLM**, a comprehensive benchmarking framework for evaluating **LLM**s in the **CeLL**ular domain. CeLLM covers a broad spectrum of tasks spanning gene, cell, and omics-level analyses, systematically assesses 15 open-source, proprietary, and biology-specialized models, and incorporates diverse evaluation criteria under multiple task settings. As a cross-scale, reproducible, and dynamic benchmark, CeLLM provides a sustainable platform to track progress, foster methodological innovation, and accelerate the development of LLMs toward virtual cell modeling.

## 1 Introduction

High-throughput single-cell sequencing technologies have profoundly advanced our ability to dissect cellular systems (Stuart & Satija, 2019; Lähnemann, 2020). They enable researchers to construct cell atlases at tissue and disease levels, supporting a wide range of downstream analyses such as cell type annotation (Stuart, 2019; Klein, 2015), trajectory inference (Trapnell, 2013; Saelens, 2019), perturbation screening (Chen, 2015), and cross-study integration (Butler, 2018; Luecken, 2022). In this context, single-cell foundation models (Cui, 2024; Zeng, 2025; Hao, 2024) have recently emerged. Typically pretrained on large-scale single-cell transcriptomic data, these models treat genes as "tokens" and cells as "sentences", thereby learning general-purpose cell representations that demonstrate potential across diverse tasks. Their advent marks a paradigm shift in single-cell analysis, from task-specific algorithms toward generalizable representation learning.

Meanwhile, the concept of the Artificial Intelligence Virtual Cell (AIVC) (Bunne, 2024) has gained increasing attention. The vision is to construct a multi-scale digital surrogate that can simulate and predict cellular behaviors in silico. Unlike models restricted to a single modality, AIVC emphasizes cross-modal integration, simultaneously absorbing information from transcriptomics, epigenomics, proteomics, metabolomics, spatial omics, and imaging data (Qian, 2025; Yang, 2025). Such integration enables modeling of molecular, cellular, and microenvironmental dynamics in a unified framework. Crucially, the core objective of AIVC is not merely to fit existing data, but to enable causal reasoning and counterfactual prediction under perturbations—for example, forecasting how cellular states evolve under drug treatment or gene editing. In addition, AIVC must be interpretable and interactive: beyond delivering predictions, it should expose its reasoning chains in a human-comprehensible manner and allow researchers to query it using natural language, thereby closing the loop between computation and experimentation. As such, AIVC is increasingly regarded as a key direction for transforming single-cell data–driven research into knowledge generation and experimental design support.

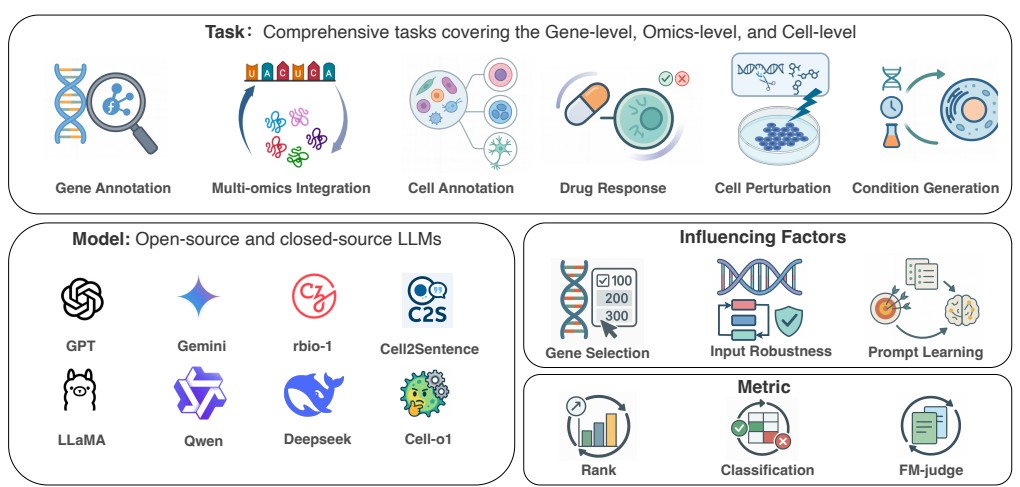

Figure 1: The overview of the CeLLM benchmark. The benchmark is organized incrementally from datasets to models, to influencing factors, to metrics, covering six major tasks and 15 open-source and closed-source models.

Despite recent progress, current single-cell foundation models remain far from fulfilling this vision. Their input layer typically relies on a single modality (e.g., scRNA-seq (Kedzierska, 2025)), limiting their ability to capture multi-omics signals; their discrete representations often lose information on expression intensity and co-expression patterns. At the task level, these models are usually optimized for a single objective, with insufficient zero-shot or few-shot generalization, requiring fine-tuning to adapt to new tasks and unable to simultaneously support multiple downstream applications within a unified system. From a modeling perspective, they largely learn correlational representations, lacking causal reasoning capabilities and thus unable to predict counterfactual outcomes under perturbations. In terms of output, most models only provide embeddings or labels, offering limited interpretability, no reasoning traces, and no natural language interface for direct scientific interaction. These limitations constrain their ability to serve as the foundation for virtual cells.

In contrast, large language models (LLMs) exhibit unique advantages that make them promising candidates toward realizing AIVC. By leveraging language as a universal representation, LLMs can map heterogeneous modalities into a shared semantic space, supporting cross-modal integration. They also demonstrate strong few-shot and zero-shot learning capabilities, enabling rapid adaptation to new tasks without extensive fine-tuning, and can handle multiple tasks under a single framework. Furthermore, LLMs internalize prior knowledge from vast literature and databases, allowing them to generate causal hypotheses about perturbations when combined with single-cell data, while producing dynamic reasoning chains in natural language. This not only enhances interpretability and usability but also facilitates interdisciplinary communication. Importantly, the knowledge coverage of LLMs extends beyond any single training dataset, enabling natural fusion of experimental data with external knowledge to support cross-domain reasoning. Notably, recent advances have shown the power of LLMs in molecular and protein sciences—for example, in small-molecule property prediction, drug generation, and protein structure (Ma, 2025; Xiao, 2025b; Liu, 2024; Xiao, 2025a) and function modeling—providing further evidence of their potential for tackling biological problems and laying the groundwork for extending their application to cellular-scale virtual modeling.

Recently, large language models (LLMs) have achieved remarkable progress in downstream tasks such as cell annotation and perturbation prediction in the field of cell biology. However, several key challenges remain, which severely limit further advances in this area:

- **Comprehensive task coverage for evaluation.** Current applications of LLMs in the cellular domain are mainly limited to tasks such as cell annotation and perturbation prediction. There is a lack of exploration for tasks such as multi-omics integration analysis, gene annotation, and both gene-level and omics-level tasks. Moreover, models applied to different

tasks have not yet been compared under a unified framework. For instance, `cell-o1` (Fang, 2025a) has been evaluated on batch-level cell annotation, but its performance on single-cell annotation remains unclear.

- **Comprehensive evaluation metrics.** While existing studies provide metrics for cell type annotation and perturbation prediction, current evaluation standards are often limited to accuracy, F1-score, and related measures. For tasks such as conditional cell generation and perturbation gene signature prediction, there is a lack of holistic evaluation metrics, making it difficult to comprehensively assess the biological relevance of model outputs.

- **Robustness to key influencing factors.** At the input level, transforming cellular expression values into textual representations suitable for LLMs can be affected by multiple factors, such as the length of selected genes and noise-induced gene ordering changes. At the downstream task level, it remains unclear whether different prompting frameworks or few-shot prompting strategies influence LLM performance across tasks, as this has not yet been systematically explored.

To address these challenges, we propose a new comprehensive LLM benchmarking framework for the cellular domain, named **CeLLM**. This benchmark unifies the evaluation of different LLMs on a wide spectrum of cellular tasks, ranging from gene annotation and cell type annotation to drug response prediction and cell perturbation prediction. CeLLM systematically evaluates 15 models, including both open-source and proprietary LLMs as well as biology-specialized foundation models, covering diverse indicators from simple classification to foundation model assessments. Furthermore, CeLLM characterizes performance under multiple task settings influenced by key factors. Designed as a cross-scale, reproducible, and future-oriented dynamic benchmark, CeLLM will continuously evolve with the emergence of new models, tasks, and datasets, serving as a long-term sustainable evaluation platform that promotes the development of LLMs toward virtual cells and lays the foundation for subsequent methodological innovations.

## 2 RELATED WORK

### 2.1 LARGE LANGUAGE MODELS FOR SCIENTIFIC PROBLEMS

In recent years, large language models (LLMs) have demonstrated broad potential in solving scientific problems across diverse domains such as mathematics, chemistry, and biomedicine. Research in this direction has primarily focused on scientific reasoning and domain adaptation. For example, OmniScience (Prabhakar, 2025) leverages domain-adaptive pretraining and instruction tuning to improve performance on scientific discovery and cross-disciplinary reasoning tasks; MedBioLM (Kim, 2025) integrates fine-tuning with retrieval-augmented generation (RAG) to enhance factual consistency and contextual understanding in medical and biomedical question answering. These studies highlight that LLMs are not only capable of processing scientific text but also hold promise for empowering complex research tasks. To systematically evaluate LLMs in more specialized scientific domains, several new benchmarks have recently been introduced. CURIE (Cui, 2025) proposes a multi-task benchmark for scientific long-context understanding and reasoning, spanning materials science, quantum computing, and protein studies, in order to assess model performance within realistic scientific workflows. PhySense (Xu, 2025) focuses on principle-based reasoning in physics, examining whether LLMs can adhere to scientific laws while generating explanatory answers. QCBench (Xie, 2025) introduces a quantitative chemistry benchmark designed to evaluate LLMs' reasoning capabilities and computational accuracy in quantitative chemical problems. In particular, at the single-cell level, how to effectively harness LLMs for data interpretation and knowledge discovery remains an open question. This study specifically targets this gap, aiming to systematically evaluate and advance the application of LLMs in cellular tasks.

### 2.2 LARGE LANGUAGE MODELS FOR CELL BIOLOGY TASKS

With the rapid progress of LLMs in natural language processing, researchers have begun to explore their potential in cell biology, particularly for addressing challenges in single-cell data analysis. For instance, Cell2Sentence (C2S) (Biderman, 2025) transforms single-cell transcriptomic data into "cell sentences", enabling LLMs to interpret and process gene expression profiles for tasks such as cell type prediction and data generation. rbio1 (Ye, 2024) leverages reinforcement learning from a

pretrained LLM, using biological world models (e.g., Gene Ontology databases) as soft verifiers to incorporate biological knowledge, thereby demonstrating the feasibility of training strong biological reasoning models with simulation-based rather than purely experimental data. In the direction of virtual cell modeling, CellForge (Chen, 2024) introduces a multi-agent framework that automates the design and optimization of virtual cell models, improving predictions of cellular responses to perturbations such as gene editing and drug treatment.cell-o1 (Fang, 2025a) reformulates cell type annotation of single-cell RNA sequencing data into a batch-level reasoning problem. PerturbQA (Wang, 2025) proposes a benchmark dedicated to perturbation experiments, covering tasks such as predicting gene expression changes, performing gene set enrichment analysis, and identifying expression directionality shifts. CellVerse (Fang, 2025b) introduces a language-driven question answering benchmark that integrates multiple single-cell datasets, aiming to evaluate LLMs' ability to operate across cell biology tasks. This benchmark highlights persistent performance gaps in handling complex perturbation and drug response prediction, underscoring both the potential and limitations of current LLMs in cellular applications. While these efforts have made notable progress in applying LLMs to cell biology, they generally suffer from limited task coverage, insufficient baseline models, incomplete evaluation metrics, and a lack of systematic exploration of key influencing factors. These challenges motivate our work, which introduces a new framework to comprehensively evaluate LLMs in cellular tasks and to advance their applications in this domain.

## 3 MODEL

In CeLLM, we evaluate three categories of models: closed-source LLMs, open-source LLMs, and domain-specific LLMs to comprehensively assess the performance of large language models.

### 3.1 CLOSED-SOURCE GENERAL LLMs

Closed-source general models, including OpenAI GPT-5 and Google Gemini 2.5, demonstrate strong generalization, long-context handling, and tool-use abilities. GPT-5 has 1.8 trillion parameters with sparse activation, using a dual-core architecture and dynamic routing to select sub-models per task. Gemini 2.5 utilizes a sparse Mixture-of-Experts (MoE) Transformer with dynamic routing to assign tokens to expert subsets, decoupling model capacity from per-token cost. Both are trained on large-scale corpora with supervised fine-tuning and RLHF, and offer reasoning modes for multi-step inference (OpenAI, 2025; Shazeer, 2025).

### 3.2 DOMAIN-SPECIFIC LLMs

Domain-specific models, including cell-o1, rBio1, and C2S, target single-cell and biological reasoning tasks. Cell-o1, built on a 7B-parameter Qwen2.5-7B-Instruct, performs batch-level scRNA-seq cell type annotation (CellPuzzles) via multi-stage training: reasoning distillation, supervised fine-tuning, and reinforcement learning, achieving joint batch reasoning and unique label assignment (Fang, 2025a). rBio1, based on a lightweight 3B-parameter Qwen2.5, combines multiple soft verifiers with RL to distill biological knowledge, enabling gene perturbation prediction and cross-cell-line generalization (Ye, 2024). C2S uses decoder-only Transformers (GPT-2 / Pythia), fine-tuned on concatenated cell sentences and natural language prompts (Biderman, 2025; Rizvi et al., 2025).

### 3.3 OPEN-SOURCE GENERAL LLMs

Open-source general-purpose models in our benchmark include DeepSeek-V3.1, the gpt-oss series, Llama-3.1-8B, and the Qwen3 series. DeepSeek-V3.1 adopts a mixture-of-experts (MoE) architecture combined with multi-level attention (MLA), low-precision computation, and hybrid inference mechanisms, leading to significant improvements over V3 in long-context modeling, tool use, and reasoning ability (Abdin, 2024). The gpt-oss series (120B/20B) employs autoregressive MoE Transformers with alternating banded and global attention, and follows a training pipeline consisting of large-scale pretraining and post-training with instruction tuning and preference optimization (OpenAI, 2025). Llama-3.1-8B, a decoder-only Transformer, extends context length through RoPE scaling and is trained in multiple stages, including pretraining, continual domain-adaptive pretraining, instruction fine-tuning, and preference alignment, thereby supporting long-text comprehension and generation (Dubey, 2025). The Qwen3 series covers a wide range of model scales, from 4B to

32B, while providing both thinking and non-thinking model variants to explicitly assess the role of reasoning in downstream tasks (Team, 2025).

## 4 CELLM TASK

In this work, we select six representative cell-level tasks to comprehensively evaluate the potential of large language models (LLMs) in realizing the vision of AI for Cell Biology (AIVC). These tasks span the genetic, multi-omics, and cellular levels.

**Gene Annotation.** The gene annotation task requires the model to describe the major functions and biological roles of a given gene. We curated a benchmark set of approximately 50 genes based on the Gene Ontology (GO) database. Since the GO database may already be present in the pretraining corpus of LLMs, this task has a certain "recall" nature. It allows us to probe whether the model has acquired fundamental molecular biology knowledge, serving as a prerequisite reference for more complex downstream tasks.

**Multi-omics Integration.** The multi-omics integration task requires the model to generate comprehensive biological interpretations or predictions by leveraging different data modalities, such as genomics, transcriptomics, epigenomics, and proteomics. This task simulates real-world research scenarios where multidimensional data must be jointly analyzed. It evaluates the ability of LLMs to perform cross-modal knowledge fusion and reasoning, thereby assessing whether they can extract patterns from heterogeneous sources and produce coherent biological insights.

**Cell Annotation.** Cell annotation is the most fundamental cell-level task. We construct datasets across three dimensions: single-cell type annotation, batch annotation, and spatial transcriptomic cell annotation. These correspond to phenotype recognition at the single-cell, batch, and spatial levels, respectively. This task systematically evaluates whether the model can correctly identify cell identity—a prerequisite for all downstream cell biology analyses.

**Drug Response Prediction.** The drug response prediction task requires the model to determine whether a cell is sensitive or resistant to a given drug. This task not only reflects whether the model can capture the relationship between drug mechanisms and cellular states but also directly maps to critical applications in precision medicine. It tests whether LLMs can transfer molecular-level knowledge into clinically relevant functional predictions.

**Cell Perturbation.** The cell perturbation task focuses on the ability of the model to reason about cellular state changes under external perturbations, such as gene knockouts. Specifically, the model must predict the impact of a perturbation on the transcriptome of a cell. The significance of this task lies in measuring whether LLMs can engage in causal reasoning about the relationships among genes, cells, and functions, rather than remaining at the level of static description.

**Conditional Cell Generation.** The conditional cell generation task requires the model to generate biologically plausible expression profiles under specified conditions, such as cell type or microenvironment. This generative and creative task evaluates whether LLMs possess the potential for design-oriented applications, such as drug development or synthetic biology. It reflects the boundary between "understanding" and "generation" in LLM-based cell biology research.

## 5 INFLUENCING FACTOR

In this section, we systematically investigate multiple factors that influence the performance of large language models (LLMs) on cell-related tasks, to assess their robustness in downstream applications. Specifically, we focus on three aspects: gene length of input, robustness to input, and prompting learning.

**Gene Length Selection.** When constructing inputs, genes are sorted by expression values in descending order, and the top $k$ gene names are sequentially provided to the model. This truncation strategy partially mimics the commonly used "feature selection" process, since highly expressed genes are often more representative and carry stronger biological signals. However, discarding low-expression genes may result in the loss of critical information. By varying the gene input length, we evaluate the sensitivity and dependency of LLMs on the amount of input information.

**Input Robustness.** Typically, LLMs take a gene list sorted from high to low expression as input. In this study, we perturb the order of genes to examine robustness: on one hand, this simulates potential noise in sequencing data; on the other, it tests whether the model has truly captured the ordering relationships among genes.

**Prompting Learning.** Prompting is a key component in the application of LLMs. We focus on the impact of few-shot prompting and prompt template design on model performance. Few-shot prompting reflects the adaptability and generalization of LLMs in low-resource scenarios, while different prompt templates help evaluate their stability under complex contexts. These factors are critical for advancing the practical deployment of LLMs in biomedical applications.

# 6 METRIC

In this section, we introduce the metrics used for CeLLM evaluation, which fall into three categories: classification metrics, ranking metrics, and foundation model–based evaluation. These three categories correspond to different task types and together provide a comprehensive view of model performance on cell-related tasks.

**Classification.** For tasks that are inherently classification problems, such as cell type annotation and drug response prediction, we adopt commonly used metrics including accuracy, recall, precision, and F1-score. In addition, we employ the Top@k overlap and ROUGE metric to evaluate the degree of overlap between predicted results and ground truth, which is particularly relevant in candidate set prediction or multi-label scenarios. These metrics directly reflect the reliability and correctness of the model in standard classification tasks.

**Rank.** For tasks that require predicting ranked outputs, such as cell perturbation prediction or cell condition generation, we employ ranking-based consistency metrics. Specifically, Spearman's rank correlation coefficient measures the overall correlation between predicted and true expression ranks, while NDCG@k emphasizes the prediction quality of the top-expressed genes. These ranking metrics assess not only whether the model captures the statistical patterns of real expression profiles, but also whether it produces biologically reasonable preferences in gene selection.

**FM-as-judge.** Relying solely on statistical measures is insufficient to capture the full quality of model outputs. Therefore, we further propose to use domain-specific foundation models (FM) as judges. In particular, we leverage a single-cell foundation model geneformer to compute the distance between predicted and true cells in biologically meaningful embedding spaces. Traditional metrics measure whether "gene orderings are similar," whereas the geneformer cosine measures whether "the generated cells are in the correct biological state." Together, they form a complementary evaluation framework that spans low-dimensional and high-dimensional spaces, combining explicit structure with implicit semantics. Moreover, for descriptive text outputs (gene annotation), we evaluate them with biomedical language models such as BioBERT (Lee, 2020) to verify their semantic plausibility and domain alignment. This class of metrics enables a holistic evaluation of LLMs, covering both biological relevance and semantic coherence.

# 7 RESULTS AND ANALYSIS

## 7.1 EXPERIMENT SETTINGS

In this section, we conduct a comprehensive experimental evaluation of large language models on cell-related tasks. The benchmark covers three levels and six categories of tasks: gene annotation, multi-omics integration, cell annotation, drug response, cell perturbation, and cell condition generation. We evaluate 15 models: `GPT-5`, `Gemini-2.5-pro`, `Gemini-2.5-flash`, `cell-o1`, `rBio1`, `C2S`, `Deepseek-V3.1`, `gpt-oss-120B`, `gpt-oss-20B`, `Llama-3.1-8B`, `Qwen3-4B`, `Qwen3-8B`, `Qwen3-14B`, `Qwen3-32B`, `Qwen3-30B-A3B`, including open-source, closed-source, and domain-specific models, spanning a variety of scales, reasoning and non-reasoning types, as well as dense and MoE architectures. The domain-specific models are deployed locally on H800 GPUs using vLLM, while the other models are accessed via their official APIs for inference. The prompt templates and dataset detail information for each task are provided in the Appendix. Due to space limitations, only the key results are presented in the main text, while full results can be found in the Appendix.

## 7.2 MAIN RESULTS

### 7.2.1 GENE ANNOTATION

**Setup.** In the gene annotation task, we require the model to recall the biological function of a gene, using GO and NCBI as reference answers, and we evaluate performance by comparing the semantic similarity between the model outputs and the reference biological texts using BioBert and ROUGE.

**Results.** From the Table 18, we observe that in the simple gene annotation task, all models perform similarly. Neither model size nor the use of reasoning appears to be a determining factor. All models demonstrate a basic level of gene comprehension.

### 7.2.2 MULTI-OMICS INTEGRATION

**Setup.** In the multi-omics task, we used CITE-seq and RNA-seq data as two different omics sources and set up three experimental conditions: CITE-seq, RNA-seq, and Integrated (using both). This design aims to investigate the performance differences of LLMs in handling single-omics versus integrated multi-omics data, thereby evaluating their cross-omics generalization ability and assessing whether they can leverage complementary information to improve the accuracy and robustness of downstream tasks.

**Results.** As summarized in Table 11, additional omics information does not lead to consistent performance improvements. In most models, the results of the Integrated and RNA settings are similar, while in some cases the Integrated setting performs much worse than RNA. The trend shows that larger models are more capable of leveraging multi-omics information, with GPT-5 and Gemini-2.5 Pro achieving better performance in the Integrated setting compared to RNA. The think mode does not provide stable improvements.

### 7.2.3 CELL ANNOTATION

**Setup.** We evaluated three levels of cell annotation: single-cell annotation, batch-level annotation, and spatial annotation. For batch-level annotation, we adopted the same metrics as cell-o1. To examine whether large language models can leverage spatial transcriptomics information, we designed comparative experiments where only single-cell information from spatial transcriptomics was provided, along with the five or ten nearest neighbors, as well as the five next-nearest neighbors, as prompt information.

**Results.** The results summarized in Table 1, Table 2, and Table 9 demonstrate that open-source models and domain-specific models exhibit capabilities comparable to closed-source models, and in some datasets even surpass them. Models trained at the batch level, such as cell-o1, show transferable performance on single-cell annotation tasks, highlighting the potential of "all-in-one" large language models for cross-level generalization. Moreover, both model size and explicit reasoning modes consistently contribute to performance improvements in cell annotation. In contrast, the C2S model, with its limited capacity, lacks the ability to effectively follow instructions. Regarding spatial information, most models tend to treat it as noise and fail to utilize it efficiently, although a subset of models can benefit from spatial transcriptomics data to achieve performance gains.

Table 1: Results of the single-cell annotation task.

| Setting | Brain | | | | Myeloid | | | | Organoid | | | |
|---|---|---|---|---|---|---|---|---|---|---|---|---|
| Metric | Precision | Recall | F1 | Accuracy | Precision | Recall | F1 | Accuracy | Precision | Recall | F1 | Accuracy |
| *Closed-source LLMs* | | | | | | | | | | | | |
| GPT-5 | 81.06 | 65.17 | 67.10 | 67.50 | 12.37 | 18.43 | 12.01 | 20.00 | **65.26** | **65.20** | **63.58** | **71.50** |
| *Domain-specific LLMs* | | | | | | | | | | | | |
| cell-o1 | 54.31 | 47.90 | 44.00 | 47.00 | 13.83 | 22.20 | 14.67 | 21.00 | 22.31 | 20.40 | 17.29 | 23.50 |
| C2S | 0.00 | 0.00 | 0.00 | 0.00 | 0.00 | 0.00 | 0.00 | 0.00 | 0.00 | 0.00 | 0.00 | 0.00 |
| *Open-source LLMs* | | | | | | | | | | | | |
| gpt-oss-120B | **81.27** | **75.31** | **76.08** | **75.50** | **19.91** | **22.25** | **18.09** | **26.00** | 62.14 | 60.53 | 58.02 | 65.00 |
| Qwen3-8B | 37.11 | 28.15 | 23.45 | 29.50 | 6.05 | 8.69 | 5.12 | 9.50 | 21.74 | 22.57 | 19.74 | 27.00 |
| Qwen3-8B-think | 52.65 | 34.75 | 30.45 | 37.00 | 10.39 | 13.80 | 8.44 | 12.00 | 37.21 | 30.61 | 24.24 | 33.00 |
| Qwen3-14B | 68.61 | 48.08 | 46.88 | 49.00 | 11.37 | 16.46 | 9.68 | 15.50 | 28.76 | 28.85 | 23.34 | 35.00 |

Table 2: Results of the spatial cell annotation task.

| Setting Metric | non | | | | top5 | | | | top10 | | | | sample5 | | | |
| --- | --- | --- | --- | --- | --- | --- | --- | --- | --- | --- | --- | --- | --- | --- | --- | --- |
| | Acc | Prec | Rec | F1 | Acc | Prec | Rec | F1 | Acc | Prec | Rec | F1 | Acc | Prec | Rec | F1 |
| *Closed-source LLMs* | | | | | | | | | | | | | | | | |
| GPT-5 | 24.00 | 27.37 | 21.29 | 21.02 | 27.00 | 33.72 | **29.00** | **25.99** | 20.00 | 30.68 | 23.27 | 21.18 | 19.00 | 25.79 | 22.82 | 19.19 |
| Gemini-2.5-pro | **32.00** | 26.76 | **30.00** | 25.54 | 23.00 | 25.96 | 23.20 | 21.17 | 23.00 | 22.78 | **27.19** | 21.58 | 29.00 | 27.50 | **28.75** | 25.47 |
| *Domain-specific LLMs* | | | | | | | | | | | | | | | | |
| rBio1 | 16.00 | 8.21 | 7.19 | 5.47 | 19.00 | 7.21 | 6.98 | 4.31 | 15.00 | 1.55 | 4.93 | 2.35 | 18.00 | 3.91 | 6.63 | 4.05 |
| *Open-source LLMs* | | | | | | | | | | | | | | | | |
| gpt-oss-120B | **32.00** | **37.17** | 24.76 | **25.58** | **30.00** | **36.15** | 19.48 | 21.84 | **35.00** | **42.62** | 25.25 | **27.63** | **33.00** | **32.42** | 22.71 | 22.74 |
| Qwen3-8B | 23.00 | 16.27 | 13.79 | 12.10 | 16.00 | 6.94 | 8.99 | 6.46 | 19.00 | 11.74 | 9.53 | 7.98 | 18.00 | 12.66 | 10.21 | 8.56 |
| Qwen3-8B-think | 23.00 | 10.52 | 14.20 | 11.07 | 22.00 | 11.17 | 12.26 | 9.83 | 24.00 | 13.25 | 13.23 | 11.56 | 24.00 | 15.78 | 12.99 | 11.89 |

### 7.2.4 CELL PERTURBATION

**Setup.** We selected the high-quality gene perturbation VCC dataset as our benchmark and designed four tasks under perturbation conditions. Specifically, (i) given background cells and expression profiles, we determine whether a specific gene is differentially expressed under a perturbation condition; (ii) if a gene is differentially expressed, we classify whether it is increase or decrease; (iii) given the control group and the perturbed highly variable gene profile, we determine whether the perturbation is caused by a specific gene. These three tasks are formulated as binary question-answering problems. (iv) Finally, given the differentially expressed gene profile of the control group, we generate the perturbed gene profile, which is formulated as a gene re-ranking task.

**Results.** From the Table 12 and Table 14, we observe that when determining differentially expressed genes or identifying perturbation genes, models tend to conservatively answer No in the absence of absolute evidence. This bias persists even when prompts explicitly indicate that 50% of the samples are positive, suggesting that the models remain overly cautious. Nevertheless, as shown in Table 13, binary classification for direct differential expression (DIR) achieves performance above random. In the perturbation cell generation task shown in Table 15, models exhibit a clear and stable scale effect, highlighting that generative tasks demand strong generalization ability and substantial knowledge capacity, while smaller models are prone to severe hallucinations. Moreover, we find that the think mode does not always lead to improved performance and can sometimes even degrade it. We hypothesize that this discrepancy arises because annotation tasks, which rely on sequential and topological reasoning over gene lists, align well with conventional chain-of-thought training, whereas differential expression tasks involve complex information analysis and querying—integrating perturbation conditions, background expression profiles, and target genes—which do not follow a purely sequential topological structure, making standard think less effective.

### 7.2.5 DRUG RESPONSE

**Setup.** We evaluated the ability of LLMs on drug response prediction, formulated as a binary classification problem to determine whether a given cell is sensitive or resistant to a drug.

**Results.** As shown in Table 16, similar to the prediction of differentially expressed genes, the models tend to predict Sensitive, and this tendency is even more pronounced under the think mode.

### 7.2.6 CELL CONDITION GENERATION

**Setup.** We evaluated the ability of LLMs to generate biologically plausible gene expression profiles conditioned on specified cell types. Models were instructed to output the top 200 most highly expressed genes in descending order, given inputs of organ origin and cell type.

**Results.** As shown in Table 17, model performance varies significantly across categories. Top-tier closed-source models (GPT-5, Gemini-2.5-pro) consistently outperform others, particularly in Cosine Similarity (Geneformer embedding) and nDCG@100, indicating superior capture of global expression patterns and ranking structure. Performance generally improves with model scale, though thinking models did not demonstrate superior performance on this task.

## 7.3 INFLUENCING FACTOR

In this section, we investigate several key factors influencing LLM performance on cellular tasks. Due to space constraints, the results are provided in Table 19, Table 20 and Table 21 of the Appendix.

**Input length selection.** We conducted an ablation study to investigate whether the length of the input gene list affects model performance. While longer lists potentially provide richer information, models may not be able to fully exploit this advantage. We evaluated gene list lengths of 20, 50, 100, 200, and 500. As shown in the Table 21, the range of 100 to 200 emerges as the optimal window for nearly all models: this length is sufficient to retain critical marker genes while avoiding excessive context length that may dilute the model's ability to focus on informative signals.

**Input robustness.** We performed random perturbations at levels of 0, 20, 40, 60, 80, and 100 with 0 corresponding to the standard brain single-cell annotation group. Here, $p\%$ denotes randomly selecting $p\%$ of the sequence and then shuffling it. We did not adopt reversal-based metrics, as perturbation scenarios in biology are more often characterized by local rank disruptions. As shown in Table 20, we found that model performance was largely consistent across all groups. Interestingly, most other models exhibited improved performance after perturbation. We hypothesize that this is due to the presence of highly expressed but non-specific genes within the same tissue in the brain dataset. These genes, appearing at the top of the gene list, may mislead models or obscure true marker genes. Our analysis confirmed this: without perturbation, models tended to focus on these non-specific genes, whereas perturbation shifted true marker genes upward in rank, allowing models to attend to them more effectively. While this phenomenon may be dataset-specific, it raises an important question: how should cell sentences be constructed to better capture cell specificity? In particular, for tissue or subtype classification, excluding highly expressed but non-marker genes may yield more representative inputs—an aspect largely overlooked by prior LLM-based cell models.

**Few-shot prompt learning.** In the single-cell annotation task on the brain dataset, we conducted a few-shot ablation experiment by placing $k = 3$ samples from the training set along with their answers into the system prompt. We designed two settings: leak (the few-shot prompt contains the cell type of the current query) and mask (the few-shot prompt does not contain the cell type of the current query). The results shown in Table 19 show that leak consistently improves performance, while mask often leads to worse results than not using few-shot at all. This may be because the model tends to rely on the provided prompt information and often treats the gene list as a set, inferring cell types by checking intersections with few-shot samples rather than performing genuine biological reasoning. When the few-shot samples include some commonly highly expressed genes, the model is prone to misclassification, and this issue cannot be effectively mitigated by prompt design.

**Prompt framing.** In the perturbed differential expression gene prediction task, we adopted two prompting frameworks: (i) asking the model whether a given gene is differentially expressed; and (ii) providing a choice space of $k = 5$ genes and asking the model to select the one most likely to be differentially expressed. We also assigned a rank to each gene based on its true differential expression level p, and computed the average rank of the model's predictions. Both tasks shared the same set of positive samples and were of equal difficulty. As shown in Table 12, in the QA setting, model responses tended to be overly conservative. Upon examining the reasoning traces, we observed that most models preferred to answer No (i.e., not differentially expressed) in the absence of decisive evidence. As a result, the recall for negative samples approached 1 across nearly all models, while the accuracy was close to random. In contrast, under the CLS setting, models demonstrated genuine reasoning ability: their accuracy substantially exceeded the random baseline of 20%, and clear scaling effects emerged, with larger models consistently performing better. The rank metric, which reflects how close the predicted gene is to the ground truth, exhibited a strong negative correlation with accuracy; predictions closer to the correct answer yielded higher accuracy.

## 8 CONCLUSION

We present **CeLLM**, a unified benchmark for evaluating large language models in cellular biology. By covering diverse tasks, models, and metrics, CeLLM provides a systematic framework to assess strengths and limitations of current approaches. Serving as a sustainable and evolving platform, CeLLM aims to accelerate the development of LLMs toward the vision of virtual cells and enable more generalizable, interpretable, and biologically grounded modeling.

ETHICS STATEMENT

We adhere to the ICLR Code of Ethics in all aspects of this research. All experimental data and results presented are authentic, with full transparency in methodology and analysis. We ensure that no form of plagiarism or falsification has occurred, and all sources are properly cited. In our work, we have taken steps to ensure fairness, equity, and respect for all contributors, avoiding any form of bias or discrimination. All contributors to this research are duly acknowledged for their roles, and any conflicts of interest have been disclosed. We have also respected the confidentiality and privacy of any sensitive data, ensuring compliance with relevant ethical guidelines. The peer review process has been conducted with the utmost respect for fairness and objectivity. Finally, we are committed to promoting an inclusive and open academic community and fostering a culture of collaboration and mutual respect.

REPRODUCIBILITY STATEMENT

We are committed to ensuring the reproducibility of our results. The code and data used in this study will be made publicly available upon publication, and we provide clear instructions for their usage to enable replication of our experiments. All hyperparameters, datasets, and experimental setups are described in detail in the paper to allow others to reproduce our findings.

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

# A  Appendix

## Use of Large Language Models

In accordance with ICLR's disclosure requirements regarding the use of large language models (LLMs), this section specifies the role and scope of LLM usage in this work:

1. **As evaluation subjects (core of the study):** This work evaluates LLMs as the primary objects of study. We design a benchmark framework for AIVC (Artificial Intelligent Virtual Cells) to assess model performance across a wide range of tasks, including multi-omics integration analysis, gene annotation, cell type annotation, conditional cell generation, and perturbation signature prediction.

2. **As a writing refinement tool (limited assistance):** During manuscript preparation, LLMs were used to provide limited assistance with grammar correction, word choice, sentence restructuring, and improving overall fluency. Importantly, LLMs were not used to draft or rewrite any core scientific content, including conceptual arguments, experimental design, data analysis, results interpretation, or figure preparation. The specific prompts and representative examples of edits have been retained by the authors and can be made available for review upon request.

3. **Accountability and verification:** All text generated or modified with LLM assistance was carefully reviewed by the authors to ensure the absence of plagiarism, fabricated information, or misleading statements. The authors take full responsibility for all content in the manuscript. LLMs are not listed as authors or contributors to this work.

## Rebuttal

### R.1  Compared with Other Benchmarks

Table 3: Task inclusion across different biological benchmarks.

| Task | Single-Cell Omics Arena | CellVerse | CeLLM |
|---|---|---|---|
| Single-Cell Annotation | ✓ | ✓ | ✓ |
| Batch-Level Cell Annotation | | | ✓ |
| Spatial Cell Annotation | | | ✓ |
| Drug Response Prediction | | ✓ | ✓ |
| Perturbation Prediction (DE & Dir) | | ✓(Only QA) | ✓(QA & CLS) |
| Perturbation Prediction (HVG) | | | ✓(QA & CLS) |
| Perturbed Cell Generation | | | ✓ |
| Gene Annotation | | | ✓ |
| Multi-Omics Integration | | | ✓ |
| Condition Cell Generation | | | ✓ |
| **Total Datasets** | **2** | **3** | **10** |

Table 4: Model and metric inclusion across different biological benchmarks.

| Benchmark | Single-Cell Omics Arena | CellVerse | CeLLM |
|---|---|---|---|
| **Model** | **8 models**: Mixtral($8\times7$B,$8\times22$B); Deepseek(67B); Qwen2(72B); Llama3(70B); GPT-4o, 4o-mini; C2S. | **12 models**: Qwen2.5(7B,32B,72B); Deepseek-V3,R1; GPT-4,4o,4o-mini,4.1,4.1-mini; C2S. | **20 models**: Qwen3(4B,7B,14B,32B,30B-A3B with thinking mode); Deepseek-V3.1; Llama(8B); GPT-oss(20B,120B); GPT5; Gemini 2.5(flash,pro); C2S; Cell-o1; rBio. |
| **Metric** | **1 type**: classification (BLEU, F1). | **1 type**: classification (Precision, Recall, F1, Accuracy). | **3 type**: rank (Spearman, NDCG@k, Rank), FM-as-judge (CosSim, BertScore), classification (Precision, Recall, F1, Accuracy, Top-k overlap, ROUGE-1). |

## R.2  OPTIONAL WINDOW AND K-SHOT STUDY

### R.2.1  OPTIONAL WINDOW

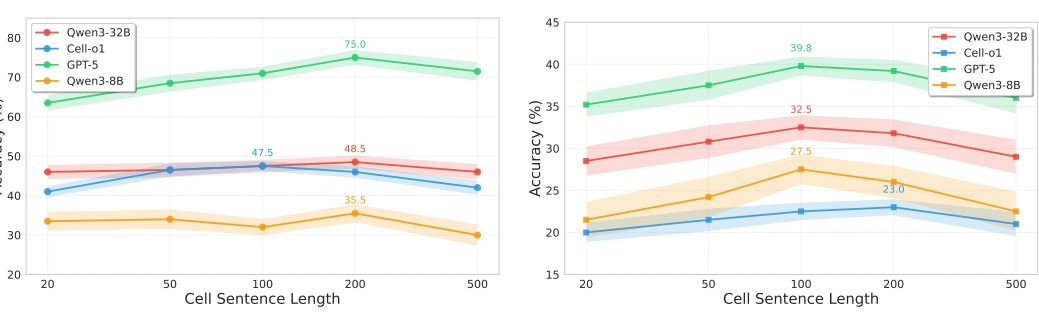

(a) single-cell annotation task.

(b) perturbation DEGs classification task.

Figure 2: Effect of varying cell sentence lengths on model performance

### R.2.2  K-SHOT WITH MASK AND LEAK STUDY

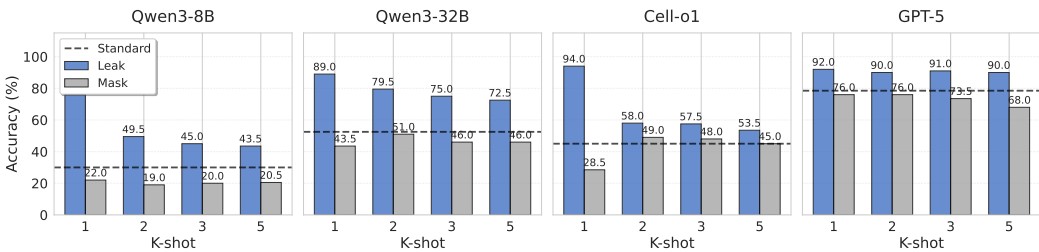

Figure 3: Performance of Mask and Leak few-shot strategies as a function of K.

## R.3  INSIGHTS AND ACTIONABLE TAKEAWAYS

Our benchmark yields the following observations and actionable recommendations:

- **Model choice should be task-dependent rather than uniformly "bigger is better".**
  - For knowledge recall tasks such as gene annotation, all models perform similarly, and frontier closed-source models provide little additional gain. Lightweight open-source models are sufficient.
  - For cell annotation tasks across single-cell, batch, and spatial settings, strong open-source models (e.g., gpt-oss-120B, Qwen3-14B) and domain-specific models (cell-o1) can match or surpass closed-source models, indicating that practitioners can rely on open models where privacy or cost is critical.
  - For generative tasks requiring full expression modeling (cell perturbation generation and conditional cell generation), model scale matters significantly: small models hallucinate heavily, whereas GPT-5 and Gemini-2.5 achieve consistently better. Frontier models are preferable when reliable generative biology is needed.

- **Reasoning ("think") modes should be used selectively.**
  - On sequential, topology-like tasks such as cell annotation with ordered gene lists, explicit reasoning modes (e.g., Qwen3 thinking models) generally improve performance.
  - On perturbation and differential expression tasks, think-mode often fails to help and can even degrade performance, likely because these tasks require integrating noisy signals rather than symbolic step-by-step reasoning.
  - **Takeaway:** Chain-of-thought benefits annotation-style reasoning over gene rankings, but should not be assumed to help all tasks; validation is needed per task family.

- **Input construction for "cell sentences" should use medium gene lengths rather than naive long contexts.**

  - Our gene list length ablation shows that 100–200 genes form an empirical "sweet spot": shorter lists lose marker information, while very long lists dilute salient signals.
  - **Takeaway:** For cell-to-text pipelines, we recommend using 100–200 genes.

- **Multi-omics should not be concatenated naïvely; larger models exploit integration more effectively.**

  - In multi-omics integration experiments, combining CITE-seq and RNA-seq into a longer prompt does not consistently improve performance, and may harm some models. Only the largest models (GPT-5, Gemini-2.5-pro) show clear benefits.
  - **Takeaway:** Multi-omics integration for LLMs is non-trivial. Structured representations or architectures are needed rather than simple concatenation.

- **Prompt framing and few-shot design critically shape model behavior.**

  - For perturbation and drug-response prediction, binary QA prompts induce a conservative bias toward "No change" or "Sensitive", resulting in near-random accuracy despite high negative recall. Framing the task as choosing among candidate genes yields much higher accuracy and clearer scale effects.
  - Few-shot prompting on brain cell annotation shows that "leaky" demonstrations (containing the target cell type) reliably improve accuracy, whereas "masked" demonstrations hurt performance. Models tend to perform gene-set pattern matching rather than pathway-level reasoning.
  - **Takeaway:** (i) Use ranking/selection-style prompts instead of yes/no QA for perturbation-style tasks; (ii) Carefully design few-shot examples to avoid label leakage and shallow gene-set matching.

- **Spatial information must be encoded carefully.**

  - In spatial cell annotation, appending nearest-neighbor gene lists is often interpreted as noise; only a few models show small gains.
  - **Takeaway:** Naive concatenation of neighbor cells is insufficient. Future work should explore structured (e.g., graph-based) or aggregated spatial encodings.

- **Evaluation design: combine classic metrics with FM-as-judge for biological realism.**

  - Traditional classification and ranking metrics capture general correctness but miss biologically meaningful differences. Embedding-based distances using single-cell foundation models (e.g., Geneformer) or biomedical text encoders (e.g., BioBERT) better capture biological and semantic fidelity.
  - **Takeaway:** Future virtual-cell models should be evaluated using both classic metrics and biologically informed embedding spaces; CeLLM provides such a protocol.

The above summarized takeaways enable both model developers and end users to directly translate CeLLM's experimental findings into concrete decisions regarding model selection, input construction, prompt design, and evaluation strategies.

**Limitations and future work.** Although CeLLM provides a systematic evaluation of foundation models across diverse single-cell and spatial transcriptomics tasks, our benchmark primarily focuses on single-turn, static model behavior. Future work will build upon CeLLM to explore cell-aware agents whose capabilities extend beyond one-shot predictions—enabling multi-step decision-making, iterative hypothesis generation and refinement, dynamic incorporation of new evidence, and autonomous biological analysis.

## R.4 DISCUSSION OF CELL REPRESENTATION IN LLMS

Our choice to use the "cell sentence" format in the benchmark is motivated by three key reasons:

**Alignment with the prevailing paradigm and benchmark focus.** The cell sentence representation is the most widely adopted format in SOTA Cell LLMs. For example, both the Cell2Sentence and

Cell-o1 models use this approach for downstream tasks. Since CeLLM aims to evaluate progress within this emerging paradigm, adopting this mainstream representation is essential.

**Information retention and interchangeability.** Although the representation is textual, it preserves critical biological information. The original C2S paper demonstrated a strong linear relationship between gene order in the cell sentence and gene expression levels in log space ($R^2$=0.85). This indicates that little information is lost during the conversion. Such interchangeability allows leveraging the strengths of LLMs in natural language processing while retaining the ability to convert the representation back into gene-expression vectors for use in traditional single-cell analysis methods.

**Robustness.** Furthermore, work (Dhodapkar, 2022) related to cell text representations highlights that the ranking mechanism serves as a form of rank normalization. This property enhances the model's robustness to noise, enabling more stable performance when processing single-cell data.

We observed that when given the input cell sentence, LLMs can automatically infer pathway-related information from the gene list in downstream task.

```
LLM Output: </analyze> The top 100 highly expressed genes in this cell include several long non-coding RNAs (
    lncRNAs) such as MALAT1, MIR99AHG, ENSG00000231557, ENSG00000225421, LINC00609, PCAT1, LINC00882, FTX,
    CASC15, LINC02899, and CASC2. These lncRNAs are often associated with regulatory functions in various
    cell types, including glial cells and neurons. Additionally, there are several genes related to ciliary
    structure and function, such as DNAH9, CFAP299, DNAH6, DNAH7, DNAH11, DNAH12, CFAP54, CFAP61, CFAP44,
    CFAP52, CFAP418-AS1, DNAI4, and HYDIN. These genes are commonly expressed in cells with motile cilia,
    such as ependymal cells and some types of epithelial cells. The presence of RORA and SOX5 suggests
    involvement in developmental processes, which is consistent with certain glial or neuronal cell types.
    NRXN1 and NCAM2 are involved in cell-cell adhesion and are commonly expressed in neurons and glia. Among
     the cell types listed, the ependymal cell is a strong candidate because of the presence of ciliary-
    related genes and the expression of RORA, SOX5, and NRXN1, all of which are relevant to ependymal cells,
     which are ciliated cells lining the ventricles of the brain and the central canal of the spinal cord.
    Furthermore, ependymal cells are known to express a variety of lncRNAs and have complex regulatory
    networks involving developmental genes. The expression profile does not strongly align with other listed
     cell types such as oligodendrocytes, astrocytes, fibroblasts, or macrophages, which have distinct
    marker genes not prominently represented here. </analyze>
<answer>ependymal cell</answer>
```

In addition, to examine other types of cell representations, we test four representative models using different variants of the cell sentence: the original version, the version augmented with expression information, and the version incorporating gene co-expression information. The results on the cell-type annotation task are shown in Table 5.

Table 5: Performance comparison(%) of different cell representations on downstream task.

| Model | cell sentence | with expression | with gene co-expression |
|---|---|---|---|
| Qwen3-8B | **72.14** | 72.04 | 66.17 |
| Qwen3-32B | **84.08** | 83.08 | 77.61 |
| Cell-o1 | **38.81** | 25.37 | 34.83 |
| GPT-5 | **89.55** | 89.55 | 89.55 |

As the table indicates, providing additional information did not yield further performance gains; on the contrary, it even negatively affected the performance of smaller models. Regarding spatial information, in the spatial cell-annotation task in our paper, we conveyed spatial context to the LLM by constructing structured multiple cell sentences based on the distances of neighboring cells to the center cell. We found that not all models were able to make effective use of this spatial information. Therefore, for spatial transcriptomics tasks, the current cell sentence format still has limitations. Designing more effective and generalizable spatial cell-encoding strategies remains an important direction for future exploration.

## R.5 DISCUSSION OF DATASET LEAKAGE

**Data provenance and novel textualization.** Except for the gene-annotation task, all CeLLM tasks are constructed directly from *primary* single-cell or spatial omics matrices (scRNA-seq, CITE-seq, spatial transcriptomics). These raw count matrices are not natural-language documents and, to the best of our knowledge, are not included in any LLM pretraining corpora. Moreover, the inputs used in CeLLM are produced through our newly designed *cell-sentence* and *multi-omics-sentence* textualization pipeline, which serializes high-dimensional omics vectors into descriptive sentences. These templates and serialization rules did not exist prior to CeLLM, making it impossible for any LLM to have encountered identical input–output pairs during pretraining.

Table 6: Summary of the CeLLM benchmark.

| Task name | Dataset | Input | Output | Metric |
|---|---|---|---|---|
| gene annotation | Collected gene GO annotation dataset | gene name | gene description | BertScore Precision, Recall, F1; ROUGE-1 Recall. |
| single-cell type annotation | Brain & Myeloid & Organoid | single cell sentence | cell type label | Precision; Recall; F1; Accuracy. |
| batch-level cell annotation | Cell Puzzle | Multi-cell sentence | cell type label list | Accuracy; Batch-level Accuracy; Uniqueness; Format. |
| spatial cell annotation | Brain1 | Multi-cell sentence | cell type label | Precision; Recall; F1; Accuracy. |
| multi-omics integration | CITE-BMMC | Multi-omics sentence | cell type label | Precision; Recall; F1; Accuracy. |
| drug response prediction | CellVerse | Drug & single cell sentence | sensitive / Resistant | Precision; Recall; F1; Accuracy. |
| perturbation prediction (DE prediction) | Virtual Cell Challenge (VCC) | single-cell sentence & perturbation gene & gene / gene candidate | Yes/No (QA); gene name (CLS) | Rank; Precision; Recall; F1; Accuracy. |
| perturbation prediction (Dir prediction) | Virtual Cell Challenge (VCC) | single-cell sentence & perturbation gene & DE gene | Yes/No | Precision; Recall; F1; Accuracy. |
| perturbation prediction with HVGS | Virtual Cell Challenge (VCC) | control group & perturbed HVG cell sentence & perturbation gene | Yes/No (QA); gene name (CLS) | Precision; Recall; F1; Accuracy. |
| perturbed cell generation | Virtual Cell Challenge (VCC) | single cell sentence(HVG) & perturbation gene | single cell sentence(HVG) | Generformer cosine similarity; Top-k overlap; NDCG@k; Spearman@k. |
| condition cell generation | Lung & Liver & Blood | condition (organ, cell type) | single cell sentence | Generformer cosine similarity; Top-k overlap; NDCG@k; Spearman@k. |

**Gene-annotation task and overlap with biological text.** For the gene-annotation task, we intentionally use Gene-Ontology–derived descriptions. These may partially overlap with public biological text, but this is by design: the goal is to evaluate whether an LLM can function as a gene-centric knowledge base. Exposure to biological corpora during pretraining is therefore part of the evaluated capability rather than an unfair advantage.

**Domain-specific models.** For domain-adapted models such as rBio, Cell-o1, and C2S, which are trained on textualized omics data, we ensure that our CeLLM test datasets do *not* overlap with the datasets used for training in their respective papers, thereby eliminating dataset-level leakage.

**Empirical evidence against contamination advantages.** Across tasks such as spatial cell annotation and differential-gene prediction, larger models do *not* consistently outperform smaller models. This pattern indicates that CeLLM tasks require genuine biological reasoning rather than memorization of any potential pretraining data, providing additional evidence that contamination is unlikely to influence our conclusions.

## R.6 SUMMARY OF CeLLM BENCHMARK

Table 6 presents a summary of the CeLLM benchmark tasks, including the datasets used, the input and output formats, and the evaluation metrics.

**Datasets, Metrics, and Prompts.** Descriptions of the datasets, evaluation metrics, and prompt templates used for the tasks in the table are provided in Appendix A.3.2, Appendix A.2, and Appendix A.5.

**Input and Output Format.** The input consists of two components: a system prompt and a user prompt. The system prompt provides task instructions, guidance, and output rules, while the user prompt contains the specific query information. In the system prompt, we require the model to place its final prediction inside answer tags so that answers can be consistently extracted and evaluated.

**Model Sampling Parameters.** LLM responses are generated using temperature = 0.7 and top_p = 0.95, following common settings that balance diversity with coherence.

**Data Processing.** CeLLM evaluates LLMs in a zero-shot setting to assess their ability to handle cell-related tasks without task-specific fine-tuning; therefore, most tasks do not contain a training set. For few-shot experiments, we split the Brain dataset into 80% training and 20% testing using a fixed random seed (42), and construct few-shot prompts using examples from the training portion. We process cellular data into core text data for various tasks according to the following steps, with a random seed of 42 applied for data sampling parts.

- **Gene Annotation**: *Structure GO terms* (Molecular Function, Biological Process, Cellular Component) and their official definitions into reference answer text.

- **Multi-omics Integration**: *Gene Normalization* is performed on scRNA-seq data; *Transcriptomics* calculates the top-200 expressed genes for each cell to generate a cell sentence; *Proteomics* extracts all surface proteins with non-zero ADT expression (max 134), ranked descendingly.

- **Single-Cell Annotation**: *Gene Name Standardization* standardizes gene names; *CellSentence Encoding* converts each sampled cell's expression profile into a top-k gene cell sentence.

- **Batch Cell Annotation**: *Batch Aggregation* calculates batch-level mean expression profiles; *Top-Genes* extracts top-k genes based on mean profiles; *Context Construction* packages batch information, top-genes, and candidate cell types into a text prompt.

- **Spatial Cell Annotation**: *Encoding* converts central and neighbor cells into top-k gene CellSentences; *Prompt Construction* concatenates these lists by distance based on task settings.

- **Drug Response Prediction**: *Gene Name Standardization* standardizes gene names; *CellSentence Encoding* converts profiles to top-k gene lists; *Prompt Construction* packages the cell sentence and drug name into a QA-style prompt.

- **Cell Perturbation**: *DE Data Sampling* uses Wilcoxon rank-sum test and Benjamini-Hochberg correction to identify significant DEGs, sampling Positive, Moderate, and Negative samples based on FDR thresholds (0.05, 0.1, 0.5); *CLS Task Construction* creates candidate sets with 1 strong positive, 2 moderate, and 2 negative samples; *Directional Sampling* categorizes significant DEGs (FDR $<$ 0.05) as Increase or Decrease based on log2 fold change; *HVG Ranking* constructs HVG lists sorted by expression values before and after perturbation.

- **Condition Cell Generation**: *Mean Profiling* calculates mean expression profiles for specific cell types in organs after log1p transformation; *Label Generation* converts these profiles into top-200 gene lists as labels.

## R.7  LLM OUTPUT ON CELL GENERATION TASK.

```
<analyze>
We need to simulate the top 200 expressed genes in **plasmacytoid dendritic cells (pDCs)** from human blood.
    These cells are specialized dendritic cells notable for strong expression of interferon-regulated genes
    (IRGs), endosomal TLRs (especially TLR7, TLR9), antigen presentation machinery, and certain surface
    receptors.
**Key features of blood pDCs:**
- **Lineage and identity markers**: IL3RA (CD123), CLEC4C (BDCA2), LILRA4 (ILT7), TCF4, SPIB.
- **Interferon response / antiviral genes**: ISG15, MX1, IFIT1, IFIT3, OAS1, OAS2, IFI6, IFI27, RSAD2 (viperin
    ).
- **Antigen presentation machinery**: HLA class II genes (HLA-DRA, HLA-DRB1, HLA-DPA1, HLA-DPB1, HLA-DQA1, HLA
    -DQB1), plus HLA class I genes (HLA-A, HLA-B, HLA-C, B2M).
- **Endosomal TLRs**: TLR7, TLR9.
- **Co-stimulatory molecules**: CD74 (invariant chain), IRF7 (master regulator of type I IFN), IRF8, etc.
- **Surface receptors involved in migration and immune sensing**: CXCR3, CXCR4, CCR7, S1PR1, ITM2C, GZMB
    sometimes reported in activated states.
- **Housekeeping / structural genes**: ACTB, GAPDH, ribosomal protein genes (RPLs, RPSs), mitochondrial genes
    (MT-CO1, MT-ND1, etc.). These tend to dominate the absolute highest expression in scRNA-seq.
Thus, the simulated list should begin with ribosomal and mitochondrial genes, B2M, actin, GAPDH, before
    transitioning into pDC-specific genes, HLA genes, interferon-stimulated genes (ISGs), and transcription
    factors. To reach 200 genes, we include a mix of: ribosomal proteins, pDC markers, HLA genes, ISGs, and
    essential metabolic genes.
</analyze>
<answer> Top 200 gene list.</answer>
```

## R.8  CASE STUDIES: ROBUSTNESS OF CELL2SENTENCE TO PROMPTS

```
Basic Question
Question:
Who are you ?
Cell2Sentence Answer:
```

```
) are expressed in the following order:
Cell 1:
MALAT1 KCNIP
```

Drug Response Prediction Task

**Question:**
Below is a list of {gene_names} ordered by descending expression level in a cell. Your task is to predict its response to the drug Erlotinib: [resistant, sensitive]. Make your choice in format 'Final Answer: resistant' or 'Final Answer: sensitive'.
{gene_names}: MALSU1 MT-TD PABPC1L FTLP1 TADA1 S100A9 FTH1P1 TXN2 RPS12P15 MT-CYB RPS16P5 RPS27AP1 RPS20 RPS14P3 RPL35AP RPL38P3 RPS15AP1 TMSB10P1 RPS5P2 RPL13AP17 RPSAP1 ATP5E RPLP1P13 NCLN EEF1A1P1 ANXA2 RPL8P2 RPL14P1 GAPDHP1 KTN1-AS1 SERPINB5 COX6CP1 HSPEP1 DSTNP4 RPL37A CAPN7 CSTA SERBP1P1 RPLP2P1 RPL36AL UBBP1 PFN1P1 TMEM259 SAP30 RPS27A RPS3A RAD51 RPL13A COX5BP6 GNB4 HNRNPA3 TMSB4XP1 CFL1P1 NDUFS7 RPS23P1 RPL36AP10 NOP14 MT-ND2 SRIP1 HSP90AB2P HIST1H4E MRPS14 VANGL2 EPS8L2 PPIH HIST1H2BE SPTBN2 RPL5P1 RECK RTN4IP1 ARHGAP21 RPL31P1 HNRNPKP1 CCDC82 RPS20P1 GCDH RPS8P3 CIR1P2 CSDE1 TALDO1 PPP1CB EIF4A3 RPL30 EIF3I RPL11P5 SLC26A4-AS1 MT-RNR2 CCNJ PTMAP2 TAX1BP3 LGALS7B RANBP1 RPS21P4 PPIC RHOC XRRA1 SRA1 RPL34P1 HINT1P1 NEFH.
**Cell2Sentence Answer:**
.
.
659__SAPA1 SPRNA9 PSMA

Batch Cell Annotation Task

**Question:**
Context: The cell is from a female at the 73-year-old stage, originating from the lung. The patient has been diagnosed with chronic obstructive pulmonary disease. The patient is a smoker. There is no cancer present.

Cell 1: MALAT1, MTCO2P22, MTATP6P29, MTND4LP30, MTND3P6, MTND2P13, MTCYBP19, MTND5P17, MTND1P22, RPL28, MIDEAS, PDE4B, SAT1, RPS2, RPS19, S100A9, RPS27, SMAP2, JUNB, CROCC, ASTN1, RPL10, S100A8, RPL37A, CD74, RPS12, RPS15, RPL18A, ELF1, RPLP2, ZNF331, RPL15, RPS3A, GLIPR2, RPL13, HOOK2, SH3BGRL3, CYBA, TXNRD1, RPS28, UPP1, ATP5MG, FTL, LPP, RPL35A, RPL13A, RPL14, TNFAIP3, SERF2, HSPBAP1
Cell 2: CCL21, MALAT1, MT2A, TMSB4X, RPL10, STEAP1B, VIM, RPL21, ASAP1, EEF1A1, RPS27, MTCO2P22, RPL32, TPT1, MTND4LP30, FTL, RHOJ, TM4SF1, CCSER1, RPLP1, RPL34, RPL13A, IL6ST, MARCHF3, MTND2P13, RPL37A, MTATP6P29, RPS14, DLC1, RPS12, RPS27A, MTCYBP19, RBFOX2, TMSB10, MTND1P22, B2M, FOS, TGFB2, RPS3A, RPS28, RPS6, ZFP36, RPL41, RPS15A, RPL39, MIR4435-2HG, RPL18A, PTMA, RPS18, RPS8
Cell 3: FTL, MTATP6P29, MTND2P13, TMSB4X, MTCO2P22, MTND4LP30, CCSER1, APOC1, MTCYBP19, MTND3P6, MT2A, B2M, RPS19, BCAS4, CCDC30, CCL18, RPS9, ADGRL3, S100A6, MTND1P22, ACTB, RPS18, MTND5P17, LGALS1, APOE, CD52, COLEC11, RPLP2, RPL34, RPL39, MT1X, RPL3, RPL27A, HSPA4, TBCA, RBMS1, IFI27, PRDX1, JPT1, R3HDM2, COX5B, NSA2, TMSB10, RPS25, S100A10, SELENOW, HLA-DRA, RPS6, S100A9, ELOB
Cell 4: MT2A, MT1E, TMSB4X, MT1X, CCSER1, MALAT1, S100A6, AGER, MTND4LP30, MT1G, CAV1, ACTB, S100A10, B2M, DSTN, RPS12, MTCO2P22, RBFOX2, RPL21, EMP2, IL6ST, RPL41, GSTP1, MTCYBP19, ATF7IP2, H3-3B, CD55, BCAS4, RPL10, RPS27, KRT7, MTATP6P29, RPL34, STEAP1B, TPT1, RPLP1, EEF1A1, RPL37, RPS18, RPS28, RPL27A, RPS16, RPLP2, RPL37A, POLR2L, KRT18, NEDD4L, FTL, MTND2P13, RPL19
Cell 5: MALAT1, MTATP6P29, RPLP1, MTCO2P22, MTND4LP30, RPL10, AREG, RPS6, B2M, RPL13, RPS24, TMSB10, RPS12, RPS27, SRGN, RPL37A, IL6ST, RPL34, MTND2P13, RPL41, RBFOX2, RPL21, RPL18A, TPT1, CD74, RPL28, RPL14, RPS8, RPL39, RPS4X, RPS15, HLA-DRA, TXNRD1, MTCYBP19, SIK3, SIPA1L1, PABPC1, RPLP2, ACTB, RPS2, RPL13A, RPS16, HLA-DRB1, RPS3A, FTL, RPL3, RPL9, RPL37, RPL19, EREG
Cell 6: MALAT1, MT2A, MTND3P6, MTCO2P22, B2M, MTND4LP30, TMSB10, MTATP6P29, MTND2P13, MTCYBP19, INSR, ADAMTS9, ACKR1, MT1E, IFITM3, IFITM2, ACTB, MT1X, LDB2, CCL14, S100A6, IGFBP7, DHFR, IL6ST, APP, RPS18, RPS17, RBFOX2, PITPNC1, ITM2B, RPL10, CLDN5, MTND1P22, HNRNPH1, FOXO1, CTNNB1, SASH1, RPL41, RPL12, RPS16, VWF, MTHFD1L, FLNB, RPS3A, RPL34, PTMA, SPARCL1, RPS19, MTND5P17, RPS15A
Cell 7: MALAT1, MTATP6P29, NEAT1, MTND3P6, MTCO2P22, MTND1P22, MTND2P13, MTND4LP30, MTCYBP19, SFTPB, MACROD2, SFTPC, PDE4D, ANK3, TANC2, MBNL1, ZNF385B, PACSIN2, DLG2, SHROOM3, MYO1D, BTBD9, MACF1, DPYD, ANXA1, USP54, LINC-PINT, CXCL2, NEDD4L, LMO7, PTPRG, DOCK5, HSPB1, TRIO, SGPP2, SYTL3, PLPP3, RAB7A, MAGI3, SNX25, FOSB, QKI, WSB1, DIAPH2, ZBTB20, ANKHD1, ACACA, LDLR, RASEF, ZSWIM6
Cell 8: MALAT1, MTATP6P29, ZSWIM6, MTCO2P22, MTCYBP19, MT2A, MTND1P22, MACF1, MACROD2, SFTPB, FHIP1A-DT, MTND2P13, VMP1, MTND3P6, IL1R1, RPLP1, RPL13, ALCAM, PALLD, DST, MTND4LP30, HNRNPH1, IL6ST, NEDD4L, RPL10, SGPP2, RBFOX2, SMYD3, APP, RPL41, RPS18, SIK3, ITGA2, EFNA5, NXN, F3_ENSG00000117525, RPS27, WWTR1, STAT3, ZFAND3, ACBD3, RPS12, TXNRD1, DENND5A, MTUS1, AOPEP, CXCL2, GCLC, SDK1, RPL35A
Cell 9: MT2A, TMSB10, ACTB, MT1M, EMP1, IGFBP7, MTCO2P22, B2M, DEPP1, MTND4LP30, HLA-B, TMSB4X, IFITM2, TM4SF1, RPL10, EPAS1, PTPRG, RPLP2, MTND2P13, ACTG1, RPL37A, STEAP1B, RPS6, TPT1, RPS8, PECAM1, IFITM3, MT1E, CCSER1, H3-3B, FTL, CD93, RPL13A, MTATP6P29, S1PR1, CAV1, MTND3P6, IL6ST, TIMP3, NES, NAMPT, RPS12, PDZD2, MTND5P17, SYN3, RPL8, RPL18, RPS27, RPL28, RPL41

Match the cells above to one of the following cell types:
CD1c-positive myeloid dendritic cell
classical monocyte
endothelial cell of lymphatic vessel
epithelial cell of lung
macrophage
malignant cell
pulmonary alveolar type 1 cell
pulmonary artery endothelial cell
vein endothelial cell

**Cell2Sentence Answer:**
of lung endothelial cell.
mucosal brick ciliated tissue group.
bas

## R.9 CASE STUDIES: WHEN LLM REASONING ALIGNS WITH KNOWN BIOLOGY?

### R.9.1 EXAMPLES OF INCORRECT ANSWERS

**True Label:** central nervous system macrophage
**LLM Output:**<analyze>The top 100 highly expressed genes in this single-cell RNA sequencing profile include a mix of long non-coding RNAs (lncRNAs) such as **MALAT1**, **NEAT1**, and **LINC00278**, which are often associated with various cellular processes, including regulation of gene expression, splicing, and chromatin remodeling. Additionally, the profile contains genes involved in a variety of cellular functions, including signaling, metabolism, and structural components. However, a closer look at the expression profile reveals some key markers that can be used to infer the cell type: 1. **QKI** (Quaking) is a well-known marker for **oligodendrocyte precursor cells (OPCs)** and **oligodendrocytes**. It is involved in RNA splicing and is highly expressed in oligodendrocyte lineage cells. 2. **MBNL1** (Muscleblind-like 1) is another marker that has been associated with oligodendrocyte precursor cells and is involved in RNA splicing and regulation of alternative splicing events. 3. **PLXND1** (Plexin D1) has been implicated in oligodendrocyte development and myelination. 4. **RUNX1** is a transcription factor involved in the regulation of oligodendrocyte differentiation and myelination. 5. **PICALM** (Phosphatidylinositol binding clathrin assembly protein) is involved in endocytosis and has been associated with oligodendrocyte lineage cells. 6. **COMMD10** is a gene that has been linked to oligodendrocyte development and myelination. These markers, particularly **QKI** and **MBNL1**, are

```
        highly specific to oligodendrocyte lineage cells, and the presence of **RUNX1** and **PICALM** further
        supports the possibility of an oligodendrocyte precursor cell or oligodendrocyte identity. While **
        oligodendrocyte** and **oligodendrocyte precursor cell** are both plausible, the expression of **RUNX1**
        and **MBNL1** is more commonly associated with **oligodendrocyte precursor cells**, which are in the
        process of differentiating into mature oligodendrocytes. Therefore, based on the expression profile and
        the presence of these specific markers, the most probable cell type is **oligodendrocyte precursor cell
        **.</analyze>
<answer>oligodendrocyte precursor cell</answer>

True Label: choroid plexus epithelial cell
LLM Output: </analyze>The given list of top 100 genes includes several markers and pathways that can help
        infer the cell type. Let's examine some of the key genes and their known associations: 1. **TRPM3**:
        This gene encodes a transient receptor potential cation channel, which is expressed in various cell
        types, including neurons and glial cells, but it is not specific to any one cell type. 2. **HTR2C**:
        This gene codes for the serotonin receptor 2C, which is expressed in neurons and some glial cells, but
        again not specific to one cell type. 3. **MALAT1**: This is a long non-coding RNA (lncRNA) known to be
        highly expressed in many cell types, including neurons and glial cells, and is involved in splicing
        regulation. 4. **RORA**: This gene is a transcription factor involved in multiple biological processes
        and is expressed in various cell types, including neurons, astrocytes, and macrophages. 5. **MSI2**:
        This gene is a marker for oligodendrocyte precursor cells (OPCs) and is highly expressed in these cells.
         6. **CFAP54**: This gene is associated with ciliary function and is expressed in various cell types,
        including some glial and epithelial cells. 7. **SLC13A4**: This gene encodes a sodium-dependent
        dicarboxylate transporter and is expressed in various tissues, including the brain. 8. **SYNE1**: This
        gene is involved in the formation of the nuclear envelope and is expressed in various cell types,
        including neurons and glial cells. 9. **ZBTB20**: This gene is a transcription factor that plays a role
        in development and is expressed in various cell types. 10. **NEAT1**: This is another lncRNA that is
        highly expressed in many cell types and is involved in nuclear organization and RNA processing. 11. **
        GPM6A**: This gene is involved in cell adhesion and is expressed in various cell types, including
        neurons and glial cells. 12. **SLC4A10**: This gene encodes a sodium bicarbonate cotransporter and is
        expressed in various tissues. 13. **DMD**: This gene is associated with Duchenne muscular dystrophy and
        is primarily expressed in skeletal muscle cells, but it is also expressed in some glial cells. 14. **
        TENM2**: This gene is involved in axon guidance and is expressed in neurons and some glial cells. 15. **
        POU2AF1**: This gene is a transcription factor that is expressed in various cell types, including
        neurons and glial cells. 16. **SLC2A12**: This gene encodes a glucose transporter and is expressed in
        various cell types, including neurons and glial cells. 17. **PLCB4**: This gene encodes a phospholipase
        C and is expressed in various cell types, including neurons and glial cells. 18. **PROS1**: This gene is
         involved in cell signaling and is expressed in various cell types, including neurons and glial cells.
        The presence of **MSI2** is a strong indicator of **oligodendrocyte precursor cells (OPCs)**, as it is a
         well-known marker for these cells. Other genes like **TRPM3**, **HTR2C**, **SYNE1**, **TENM2**, and **
        SLC2A12** are also expressed in OPCs or related cell types. The overall expression profile suggests a
        cell with characteristics of glial cells, particularly those involved in myelination and development.
        Considering all of this, the most probable cell type is **oligodendrocyte precursor cell**.</analyze>
<answer>oligodendrocyte precursor cell</answer>
```

## R.9.2 Examples of correct answers

```
True Label: oligodendrocyte
LLM Output: <analyze>The given list of top 100 highly expressed genes provides a strong indication of the cell
        type based on the presence of known marker genes and functional pathways. Several of the top genes in
        this list are well-established markers of oligodendrocytes, the myelinating cells of the central nervous
         system. For example: MBP (Myelin Basic Protein) is a hallmark gene of oligodendrocytes, essential for
        myelin sheath formation. PLP1 (Proteolipid protein 1) is another critical myelin component,
        predominantly expressed in oligodendrocytes. QKI (Quaking) is a RNA-binding protein involved in
        myelination and is specifically expressed in oligodendrocytes. PCDH9 (Protocadherin 9) is also
        associated with oligodendrocytes and myelination. CTNNA3 (Catenin alpha 3) is involved in cell adhesion
        and has been linked to oligodendrocyte development. SLC24A2 (Solute carrier family 24 member 2) is
        implicated in oligodendrocyte function and myelin maintenance. Other genes like DST (Dystonin), SLC44A1,
         CDH11, CNTNAP2, and RTN4 (Nogo-A) are also known to be expressed in oligodendrocytes or are involved in
         processes related to myelination or axon-glial interactions. These genes are not typically expressed at
         high levels in other cell types such as astrocytes, macrophages, fibroblasts, or endothelial cells. The
         expression profile is highly specific to oligodendrocytes, making this the most likely cell type.</
        analyze>
<answer>oligodendrocyte</answer>

True Label: ependymal cell
LLM Output: </analyze> The top 100 highly expressed genes in this cell include several long non-coding RNAs (
        lncRNAs) such as MALAT1, MIR99AHG, ENSG00000231557, ENSG00000225421, LINC00603, PCAT1, LINC00882, FTX,
        CASC15, LINC02899, and CASC2. These lncRNAs are often associated with regulatory functions in various
        cell types, including glial cells and neurons. Additionally, there are several genes related to ciliary
        structure and function, such as DNAH9, CFAP299, DNAH6, DNAH7, DNAH11, DNAH12, CFAP54, CFAP61, CFAP44,
        CFAP52, CFAP418-AS1, DNAI4, and HYDIN. These genes are commonly expressed in cells with motile cilia,
        such as ependymal cells and some types of epithelial cells. The presence of RORA and SOX5 suggests
        involvement in developmental processes, which is consistent with certain glial or neuronal cell types.
        NRXN1 and NCAM2 are involved in cell-cell adhesion and are commonly expressed in neurons and glia. Among
         the cell types listed, ependymal cell is a strong candidate because of the presence of ciliary-related
        genes and the expression of RORA, SOX5, and NRXN1, all of which are relevant to ependymal cells, which
        are ciliated cells lining the ventricles of the brain and the central canal of the spinal cord.
        Furthermore, ependymal cells are known to express a variety of lncRNAs and have complex regulatory
        networks involving developmental genes. The expression profile does not strongly align with other listed
         cell types such as oligodendrocytes, astrocytes, fibroblasts, or macrophages, which have distinct
        marker genes not prominently represented here. </analyze>
<answer>ependymal cell</answer>
```

## A.1 MODELS

Table 7: The release dates and sources of the evaluated models.

| Model | Release Date | Source |
|-------|--------------|--------|
| *Closed-source LLMs* | | |
| GPT-5 | 2025-08 | https://platform.openai.com/docs/models/gpt-5 |
| Gemini-2.5-pro | 2025-06 | https://ai.google.dev/gemini-api/docs/models#gemini-2.5-pro |
| Gemini-2.5-flash | 2025-06 | https://ai.google.dev/gemini-api/docs/models#gemini-2.5-flash |
| *Domain-specific LLMs* | | |
| cell-o1 | 2025-06 | https://huggingface.co/ncbi/Cell-o1 |
| rBio1 | 2025-08 | https://virtualcellmodels.cziscience.com/model/rbio |
| C2S | 2025-04 | https://huggingface.co/vandijklab/C2S-Scale-Pythia-1b-pt |
| *Open-source LLMs* | | |
| Deepseek-V3.1 | 2025-09 | https://huggingface.co/deepseek-ai/DeepSeek-V3.1 |
| gpt-oss-120B | 2025-08 | https://huggingface.co/openai/gpt-oss-120b |
| gpt-oss-20B | 2025-08 | https://huggingface.co/openai/gpt-oss-20b |
| Llama-3.1-8B | 2024-12 | https://huggingface.co/meta-llama/Llama-3.1-8B |
| Qwen3-4B | 2025-08 | https://huggingface.co/Qwen/Qwen3-4B |
| Qwen3-8B | 2025-08 | https://huggingface.co/Qwen/Qwen3-8B |
| Qwen3-14B | 2025-08 | https://huggingface.co/Qwen/Qwen3-14B |
| Qwen3-30B-A3B | 2025-08 | https://huggingface.co/Qwen/Qwen3-30B-A3B |
| Qwen3-32B | 2025-08 | https://huggingface.co/Qwen/Qwen3-32B |

## A.2 METRICS

In this appendix, we provide the formal definitions of all evaluation metrics used for CeLLM evaluation, corresponding to the three categories in the main text: classification, ranking, and foundation model–based evaluation.

### A.2.1 CLASSIFICATION METRICS

For classification-style tasks (e.g., cell type annotation and drug response prediction), we adopt standard metrics. Let TP, FP, TN, and FN denote true positives, false positives, true negatives, and false negatives, respectively.

$$\text{Precision} = \frac{\text{TP}}{\text{TP} + \text{FP}}, \tag{1}$$

$$\text{Recall} = \frac{\text{TP}}{\text{TP} + \text{FN}}, \tag{2}$$

$$\text{F1-score} = \frac{2 \cdot \text{Precision} \cdot \text{Recall}}{\text{Precision} + \text{Recall}}, \tag{3}$$

$$\text{Accuracy} = \frac{\text{TP} + \text{TN}}{\text{TP} + \text{FP} + \text{TN} + \text{FN}}. \tag{4}$$

**Top-$k$ overlap.** For multi-label or candidate set prediction tasks, we compute the overlap between predicted and reference sets:

$$\text{Top-}k = \frac{|\text{Top}_k(\text{pred}) \cap \text{Top}_k(\text{ref})|}{k}. \tag{5}$$

**ROUGE-1 recall.** Given a reference set $R$ and a predicted set $P$, ROUGE-1 recall is defined as:

$$\text{ROUGE-1 Recall} = \frac{|\text{Unigrams}(R) \cap \text{Unigrams}(P)|}{|\text{Unigrams}(R)|}. \tag{6}$$

**Batch-level accuracy.** The proportion of batches where all predicted labels exactly match the ground truth.

### A.2.2 RANKING METRICS

For tasks requiring ranked outputs, such as perturbation prediction or condition generation, we adopt the following.

**Spearman's rank correlation ($\rho$).** For $n$ items, let $r_i$ and $s_i$ denote the ranks of item $i$ in the predicted and reference lists:

$$\rho = 1 - \frac{6 \sum_{i=1}^{n} (r_i - s_i)^2}{n(n^2 - 1)}. \tag{7}$$

**Normalized Discounted Cumulative Gain (NDCG@k).** Given graded relevance scores $rel_i$ at rank position $i$, the DCG@k and NDCG@k are:

$$\text{DCG@}k = \sum_{i=1}^{k} \frac{2^{rel_i} - 1}{\log_2(i+1)}, \tag{8}$$

$$\text{NDCG@}k = \frac{\text{DCG@}k}{\text{IDCG@}k}, \tag{9}$$

where IDCG@$k$ denotes the maximum achievable DCG@k (ideal ranking).

In our experiments, we instantiate relevance using logarithmic gain: $rel_i = 1/\log_2(1 + r_i)$, where $r_i$ is the ground-truth rank of the item at position $i$ (or 0 if unranked).

**Rank.** The Rank metric measures the average ranking of the model's predicted answer within the candidate list for differential expression gene classification. A lower value is better, indicating that the model tends to select ground truth answers with higher rankings (i.e., genes more likely to be differentially expressed).

### A.2.3 FOUNDATION MODEL−BASED EVALUATION

Beyond statistical consistency, we also employ domain-specific foundation models (FMs) as evaluators.

**Geneformer-based cosine similarity.** Given a cell-sentence gene list as input (with genes not in the Geneformer vocabulary filtered out), we extract the CLS token embedding from the final Transformer layer of the Geneformer model and use it as the cell representation $\mathbf{z}$. Let $\mathbf{z}_{\text{pred}} \in \mathbb{R}^d$ and $\mathbf{z}_{\text{ref}} \in \mathbb{R}^d$ denote the embeddings of the predicted and reference cells obtained from a pretrained geneformer. We compute the cosine similarity:

$$\text{CosSim}(\mathbf{z}_{\text{pred}}, \mathbf{z}_{\text{ref}}) = \frac{\mathbf{z}_{\text{pred}}^{\top} \mathbf{z}_{\text{ref}}}{\|\mathbf{z}_{\text{pred}}\| \, \|\mathbf{z}_{\text{ref}}\|}. \tag{10}$$

This metric quantifies the biological alignment of predictions in the latent space.

**BERTScore with BioBERT.** For textual outputs (e.g., gene function annotations), we use BioBERT embeddings (Lee, 2020). Let $\mathbf{x} = (x_1, \ldots, x_m)$ be the reference token embeddings and $\mathbf{y} = (y_1, \ldots, y_n)$ the prediction embeddings. The pairwise cosine similarity is

$$S_{ij} = \frac{x_i^{\top} y_j}{\|x_i\| \, \|y_j\|}. \tag{11}$$

The BERTScore calculation adopted in this paper is as follows:

$$\text{BERTScore-Recall} = \frac{1}{m} \sum_{i=1}^{m} \max_j S_{ij}, \quad \text{BERTScore} - \text{Precision} = \frac{1}{n} \sum_{j=1}^{n} \max_i S_{ij}, \tag{12}$$

$$\text{BERTScore} - \text{F1} = 2 \cdot \frac{\text{BERTScore} - \text{Precision} \cdot \text{BERTScore} - \text{Recall}}{\text{BERTScore} - \text{Precision} + \text{BERTScore} - \text{Recall}}. \tag{13}$$

### A.2.4 OTHER METRICS

**Format Validity.** The proportion of outputs that follow the required response format.

**Answer Uniqueness.** The average proportion of unique cell type predictions per batch.

This family of metrics enables a holistic evaluation by combining correctness, ranking consistency, and biological or semantic plausibility.

## A.3 DATASETS

This section introduces the datasets used in CELLM and their processing workflow. We first describe the global gene symbol standardization, then outline the reference datasets, and finally present the construction of task-specific text datasets.

### A.3.1 GENE SYMBOL STANDARDIZATION

To ensure cross-dataset consistency, we established a unified gene symbol mapping. Official mappings from **GENCODE** and **HGNC** (Ensembl Gene ID → HGNC Symbol) were merged, followed by deduplication, case normalization, alias resolution, and one-to-many conflict handling, where only the primary symbol was retained. Importantly, many raw datasets contain *previous symbols* and *alias symbols*, which can lead to ambiguities or even gene duplication across studies. Our standardization pipeline resolves these issues by mapping all variants to the authoritative **HGNC primary symbol**, thereby producing a clean and human-readable representation of gene names. All datasets were standardized prior to downstream processing; unmapped genes were preserved but not discarded. Compared to earlier approaches (which did not strictly standardize gene symbols), this procedure substantially reduces noise and ensures the construction of text samples that are maximally suitable for LLMs (Frankish et al., 2025; Tweedie et al., 2021).

### A.3.2 REFERENCE DATASETS

We employed a diverse set of authoritative resources covering single-cell, spatial, multi-omics, and perturbation contexts:

- **Virtual Cell Challenge (VCC) (cell perturbation)**: The dataset(Roohani et al., 2025) was generated using single-cell functional genomics, producing approximately 300,000 single-cell RNA-seq profiles by silencing 300 carefully selected genes using CRISPR interference (CRISPRi). To obtain the single-cell gene expression profiles, 10x Genomics GEM-X Flex and Illumina sequencing were utilized.

- **Gene annotation dataset(gene Annotation)**:The benchmark dataset comprises 50 genes annotations. These genes annotations were built based on a search of the GO database to test the model's fundamental biological knowledge.

- **Brain(single-cell annotation)**: BICCN/Allen whole-brain atlas (MERFISH & scRNA-seq) (Yao, 2023; Zhang, 2023).

- **Myeloid (single-cell annotation)**: 10x PBMC 68k (Donor A) (10x Genomics, 2017; Zheng, 2017).

- **Organoid (single-cell annotation)**: Human brain organoid scRNA-seq datasets (Velasco, 2019; He, 2024).

- **CellPuzzles (batch-level annotation)**: A multi-batch benchmark built on CELLXGENE Discover/Census resources (Yang & Gomez, 2024; cel, 2025).

- **Brain1 (spatial cell annotation)**: Brain slices from the STOFM benchmark (Zhao et al., 2025).

- **CellVerse (drug response prediction)**: A benchmark suite of cell/drug tasks; we use the Drug Response Prediction subset (Fang, 2025b).

- **Lung / Liver / Blood ( condition cell generation)**: HLCA (lung), Aizarani liver atlas (liver), and PBMC 68k (blood) (Sikkema, 2023; Aizarani, 2019; 10x Genomics, 2017).

- **CITE-BMMC (multi-omics integration)**: OpenProblems multimodal benchmark (Luecken, 2021; 2025).

### A.3.3 Task-specific Text Data Generation

Based on the standardized symbols and reference datasets, we unify diverse tasks into textual input–output interfaces, with specific evaluation objectives and metrics.

**Cell Annotation.** *Goal:* Assign biological cell types to single-cell transcriptomes. *Single-cell annotation*: Brain, Myeloid, and Organoid datasets were transformed into gene-ranked lists ("gene sentences"), with hierarchical cell-type labels as supervision (Yao, 2023; Zhang, 2023; 10x Genomics, 2017; Zheng, 2017; Velasco, 2019; He, 2024). Metrics include **accuracy**, **recall**, **batch-level accuracy**, **uniqueness**, and **response length**, with optional text metrics (ROUGE, BERTScore). *Batch-level annotation*: Using CellPuzzles, we aggregate top-genes and metadata within each batch, requiring globally consistent batch-level predictions; report both cell- and batch-level metrics (Yang & Gomez, 2024; cel, 2025). *Spatial annotation*: Using Brain1 (STOFM), each cell/spot is represented by a gene sentence augmented with absolute coordinates and neighborhood graphs; evaluate accuracy/recall and spatial consistency.

**Drug Response.** *Goal:* Predict cellular response to drug perturbations or answer drug-related queries. Based on CellVerse DRP, inputs combine drug/perturbation context with cell samples, rewritten as QA-style or classification prompts; outputs are categorical responses or explanatory text (Fang, 2025b). Metrics: **accuracy**, **recall** (classification); **ROUGE**, **BERTScore** (text).

**Gene Annotation.** *Goal:* Explain gene function using authoritative knowledge bases. We employ Gene Ontology (GO) and NCBI resources (AmiGO portal; NCBI Gene; GEO). Inputs are gene lists or queries; outputs are functional categories, term definitions, or enrichment results. Metrics: text similarity (**ROUGE**, **BERTScore**) and structured matching (strict/soft) (The Gene Ontology Consortium, 2023; Carbon et al., 2009; Sayers et al., 2024; Clough et al., 2024).

**Cell Generation.** *Goal:* Learn conditional generation of cell expression profiles. Using HLCA (lung), Aizarani liver atlas, and PBMC 68k, inputs are (organ, cell type, condition) and outputs are bulk-like mean profiles rendered as ranked gene lists. Metrics: **top-$k$ overlap**, **Spearman correlation**, **nDCG**, **cosine similarity**, plus optional distributional metrics (KS, EMD) (Sikkema, 2023; Aizarani, 2019; 10x Genomics, 2017).

**Perturbation Prediction.** *Goal:* Model perturbation outcomes with curated perturbation resources. We consider four sub-tasks: (i) *DEG prediction*, (ii) *direction prediction* (up/down), (iii) *condition prediction*, and (iv) *perturbed cell generation*. Inputs are (cell type + perturbation); outputs are predictions or generated profiles. We reference SCPERTURB and the NeurIPS OP3 perturbation challenge as representative sources for data/task design (Peidli, 2024; Szałata, 2024). Metrics include ranking (**Spearman@$k$**, **nDCG**, **top-$k$ overlap**, **cosine similarity**) and classification (**accuracy**, **recall**).

**Multi-omics Integration.** *Goal:* Perform cross-modality prediction, matching, and joint embedding while correcting batch effects. Using OpenProblems, tasks include (i) *modality prediction* (RNA $\leftrightarrow$ ADT/ATAC), (ii) *modality matching*, and (iii) *joint embedding*. Evaluation covers prediction accuracy/correlation, matching precision/recall, and embedding quality (structure preservation and batch removal) (Luecken, 2021; 2025).

A.4 EXPERIMENTAL RESULTS

Table 8: Results of the single cell annotation task.

| Setting | Brain | | | | Myeloid | | | | Organoid | | | |
|---|---|---|---|---|---|---|---|---|---|---|---|---|
| Metric | Precision | Recall | F1 | Accuracy | Precision | Recall | F1 | Accuracy | Precision | Recall | F1 | Accuracy |
| *Closed-source LLMs* | | | | | | | | | | | | |
| GPT-5 | 81.06 | 65.17 | 67.10 | 67.50 | 12.37 | 18.43 | 12.01 | 20.00 | 65.26 | **65.20** | **63.58** | **71.50** |
| Gemini-2.5-pro | 80.98 | 65.57 | 66.61 | 67.50 | 17.12 | 20.73 | 13.67 | 20.00 | **65.94** | 63.87 | 62.57 | 66.00 |
| Gemini-2.5-flash | 76.16 | 45.57 | 50.13 | 49.00 | 13.97 | 16.28 | 11.36 | 19.00 | 61.92 | 44.85 | 46.10 | 48.00 |
| *Domain-specific LLMs* | | | | | | | | | | | | |
| cell-o1 | 54.31 | 47.90 | 44.00 | 47.00 | 13.83 | **22.20** | 14.67 | 21.00 | 22.31 | 20.40 | 17.29 | 23.50 |
| rBio1 | 18.31 | 16.72 | 14.25 | 16.00 | 1.45 | 4.80 | 1.97 | 5.50 | 11.15 | 13.81 | 11.93 | 15.00 |
| C2S | 0.00 | 0.00 | 0.00 | 0.00 | 0.00 | 0.00 | 0.00 | 0.00 | 0.00 | 0.00 | 0.00 | 0.00 |
| *Open-source LLMs* | | | | | | | | | | | | |
| Deepseek-V3.1 | 64.51 | 19.90 | 25.77 | 20.50 | **22.77** | 8.52 | 10.28 | 10.00 | 52.88 | 22.58 | 30.97 | 24.50 |
| gpt-oss-120B | 81.27 | **75.31** | **76.08** | **75.50** | 19.91 | 22.25 | **18.09** | **26.00** | 62.14 | 60.53 | 58.02 | 65.00 |
| gpt-oss-20B | **86.16** | 39.33 | 50.51 | 41.00 | 22.47 | 18.05 | 16.49 | 20.50 | 60.57 | 38.21 | 44.61 | 41.00 |
| Llama-3.1-8B | 40.73 | 20.55 | 19.38 | 20.00 | 2.73 | 6.42 | 3.37 | 7.00 | 24.62 | 21.16 | 18.04 | 25.00 |
| Qwen3-4B | 37.10 | 21.73 | 16.69 | 22.50 | 10.34 | 4.44 | 3.62 | 5.00 | 15.85 | 14.71 | 12.89 | 17.00 |
| Qwen3-4B-think | 43.12 | 32.60 | 26.88 | 34.00 | 4.60 | 8.60 | 3.01 | 9.00 | 21.66 | 19.52 | 18.27 | 22.50 |
| Qwen3-8B | 37.11 | 28.15 | 23.45 | 29.50 | 6.05 | 8.69 | 5.12 | 9.50 | 21.74 | 22.57 | 19.74 | 27.00 |
| Qwen3-8B-think | 52.65 | 34.75 | 30.45 | 37.00 | 10.39 | 13.80 | 8.44 | 12.00 | 37.21 | 30.61 | 24.24 | 33.00 |
| Qwen3-14B | 68.61 | 48.08 | 46.88 | 49.00 | 11.37 | 16.46 | 9.68 | 15.50 | 28.76 | 28.85 | 23.34 | 35.00 |
| Qwen3-14B-think | 70.06 | 56.40 | 55.34 | 57.50 | 13.46 | 20.09 | 11.96 | 19.00 | 43.34 | 38.77 | 35.67 | 44.50 |
| Qwen3-30B-A3B | 49.90 | 35.50 | 33.70 | 37.50 | 12.30 | 12.28 | 9.37 | 13.50 | 26.64 | 29.09 | 22.75 | 35.50 |
| Qwen3-30B-A3B-think | 60.33 | 49.45 | 48.15 | 52.00 | 19.12 | 18.06 | 12.95 | 20.00 | 34.13 | 31.75 | 25.55 | 38.50 |
| Qwen3-32B | 51.70 | 43.96 | 42.45 | 45.00 | 8.84 | 13.37 | 9.30 | 15.00 | 31.04 | 31.77 | 28.43 | 37.50 |
| Qwen3-32B-think | 62.72 | 56.27 | 53.70 | 58.00 | 11.12 | 14.84 | 9.69 | 16.50 | 31.99 | 35.98 | 30.91 | 43.00 |

Table 9: Results of the cell puzzles task.

| Model | Cell-level Acc | Batch-level Acc | Uniqueness | Format |
|---|---|---|---|---|
| *Closed-source LLMs* | | | | |
| GPT-5 | 63.04 | 14.16 | **100.00** | **100.00** |
| Gemini-2.5-pro | 66.35 | 20.64 | 97.17 | 97.17 |
| Gemini-2.5-flash | 34.20 | 7.12 | 81.05 | 81.10 |
| *Domain-specific LLMs* | | | | |
| cell-o1 | **71.84** | **31.14** | 99.88 | **100.00** |
| rBio-1-3B | 6.28 | 0.00 | 52.24 | 82.28 |
| C2S-1B | 0.00 | 0.00 | 0.00 | 0.00 |
| *Open-source LLMs* | | | | |
| Deepseek-V3.1 | 58.85 | 15.16 | 92.24 | 92.33 |
| gpt-oss-120B | 60.66 | 18.08 | 94.77 | 94.79 |
| gpt-oss-20B | 2.13 | 0.27 | 4.46 | 4.47 |
| Llama-3.1-8B | 2.70 | 0.00 | 7.05 | 15.71 |
| Qwen3-4B-think | 23.82 | 0.09 | 71.15 | 97.81 |
| Qwen3-8B-think | 31.21 | 1.37 | 83.71 | 99.63 |
| Qwen3-30B-A3B-think | 37.07 | 1.74 | 78.65 | 96.71 |
| Qwen3-32B-think | 42.68 | 4.02 | 94.75 | 99.73 |

Table 10: Results of the spatial cell annotation task.

| Setting | non | | | | top5 | | | | top10 | | | | sample5 | | | |
|---|---|---|---|---|---|---|---|---|---|---|---|---|---|---|---|---|
| Metric | Acc | Prec | Rec | F1 | Acc | Prec | Rec | F1 | Acc | Prec | Rec | F1 | Acc | Prec | Rec | F1 |
| *Closed-source LLMs* | | | | | | | | | | | | | | | | |
| GPT-5 | 24.00 | 27.37 | 21.29 | 21.02 | 27.00 | 33.72 | 29.00 | **25.99** | 20.00 | 30.68 | 23.27 | 21.18 | 19.00 | 25.79 | 22.82 | 19.19 |
| Gemini-2.5-pro | 32.00 | 26.76 | 30.00 | 25.54 | 23.00 | 25.96 | 23.20 | 21.17 | 23.00 | 22.78 | 27.19 | 21.58 | 29.00 | 27.50 | **28.75** | **25.47** |
| Gemini-2.5-flash | 27.00 | 24.06 | 26.01 | 23.26 | 10.00 | 19.00 | 11.96 | 13.40 | 10.00 | 21.67 | 8.94 | 10.85 | 12.00 | 25.55 | 11.36 | 13.72 |
| *Domain-specific LLMs* | | | | | | | | | | | | | | | | |
| cell-o1 | 11.00 | 14.45 | 8.23 | 9.11 | 6.00 | 7.09 | 8.82 | 4.54 | 7.00 | 4.70 | 4.41 | 4.21 | 10.00 | 10.14 | 6.44 | 6.39 |
| rBio1 | 16.00 | 8.21 | 7.19 | 5.47 | 19.00 | 7.21 | 6.98 | 4.31 | 15.00 | 1.55 | 4.93 | 2.35 | 18.00 | 3.91 | 6.63 | 4.05 |
| C2S | 0.00 | 0.00 | 0.00 | 0.00 | 0.00 | 0.00 | 0.00 | 0.00 | 0.00 | 0.00 | 0.00 | 0.00 | 0.00 | 0.00 | 0.00 | 0.00 |
| *Open-source LLMs* | | | | | | | | | | | | | | | | |
| Deepseek-V3.1 | 12.00 | 18.41 | 10.90 | 12.85 | 11.00 | 21.77 | 10.76 | 12.84 | 7.00 | 15.92 | 8.98 | 7.44 | 12.00 | 20.49 | 7.87 | 10.34 |
| gpt-oss-120B | 32.00 | **37.17** | 24.76 | **25.58** | 30.00 | **36.15** | 19.48 | 21.84 | 35.00 | 42.62 | 25.25 | 27.63 | **33.00** | **32.42** | 22.71 | 22.74 |
| gpt-oss-20B | 21.00 | 28.95 | 14.38 | 18.21 | 12.00 | 11.97 | 7.26 | 8.01 | 22.00 | 30.85 | 14.48 | 18.54 | 18.00 | 23.52 | 12.78 | 16.04 |
| Llama-3.1-8B | 11.00 | 3.10 | 3.96 | 3.07 | 14.00 | 5.53 | 7.43 | 5.08 | 15.00 | 9.19 | 10.42 | 8.30 | 11.00 | 5.17 | 6.49 | 4.53 |
| Qwen3-4B | 22.00 | 18.03 | 12.69 | 10.25 | 18.00 | 4.58 | 7.85 | 4.44 | 13.00 | 11.50 | 6.77 | 6.63 | 16.00 | 6.77 | 6.49 | 4.12 |
| Qwen3-4B-think | 31.00 | 19.88 | 16.89 | 13.95 | 22.00 | 14.52 | 10.47 | 8.73 | 11.00 | 6.70 | 4.44 | 3.74 | 15.00 | 6.00 | 6.88 | 5.43 |
| Qwen3-8B | 23.00 | 16.27 | 13.79 | 12.10 | 16.00 | 6.94 | 8.99 | 6.46 | 19.00 | 11.74 | 9.53 | 7.98 | 18.00 | 12.66 | 10.21 | 8.56 |
| Qwen3-8B-think | 23.00 | 10.52 | 14.20 | 11.07 | 22.00 | 11.17 | 12.26 | 9.83 | 24.00 | 13.25 | 13.23 | 11.56 | 24.00 | 15.78 | 12.99 | 11.89 |
| Qwen3-30B-A3B | 25.00 | 15.48 | 17.62 | 13.46 | 20.00 | 8.95 | 11.18 | 9.22 | 20.00 | 16.77 | 14.88 | 12.69 | 25.00 | 22.14 | 16.65 | 16.25 |
| Qwen3-30B-A3B-think | **33.00** | 21.57 | **27.42** | 20.29 | 32.00 | 17.44 | 22.05 | 17.21 | 27.00 | 18.68 | 20.33 | 16.81 | 30.00 | 19.84 | 22.31 | 18.33 |
| Qwen3-32B | 29.00 | 23.28 | 23.34 | 21.64 | 23.00 | 19.88 | 16.29 | 15.67 | 25.00 | 18.98 | 18.91 | 16.79 | 20.00 | 19.69 | 22.78 | 17.21 |
| Qwen3-32B-think | 32.00 | 27.56 | 24.66 | 20.76 | **33.00** | 21.78 | **24.45** | 20.00 | 31.00 | 21.67 | 21.08 | 18.49 | 31.00 | 19.65 | 22.05 | 18.22 |

Table 11: Results of the multi-omics cell annotation task.

| Setting | scCite | | | | scRNA | | | | Integrated | | | |
|---|---|---|---|---|---|---|---|---|---|---|---|---|
| Metric | Precision | Recall | F1 | Accuracy | Precision | Recall | F1 | Accuracy | Precision | Recall | F1 | Accuracy |
| *Closed-source LLMs* | | | | | | | | | | | | |
| GPT-5 | 16.90 | 16.90 | 11.58 | 18.50 | 19.37 | 23.72 | 18.95 | 48.00 | 29.88 | 31.96 | 26.71 | 49.00 |
| Gemini-2.5-pro | 30.39 | **30.65** | 26.35 | **33.00** | **31.27** | 24.60 | 24.84 | 45.50 | **39.05** | **34.68** | **34.05** | **53.50** |
| Gemini-2.5-flash | **32.50** | 15.21 | 18.86 | 12.50 | 24.92 | 12.47 | 15.22 | 24.00 | 29.49 | 17.79 | 19.46 | 23.50 |
| *Domain-specific LLMs* | | | | | | | | | | | | |
| cell-o1 | 1.81 | 1.01 | 0.70 | 4.00 | 14.27 | 18.50 | 14.81 | 42.50 | 7.97 | 7.95 | 7.32 | 24.00 |
| rBio1 | 2.23 | 3.87 | 2.11 | 7.00 | 6.77 | 6.04 | 3.72 | 9.50 | 3.41 | 4.42 | 3.06 | 8.50 |
| C2S | 0.00 | 0.00 | 0.00 | 0.00 | 0.00 | 0.00 | 0.00 | 0.00 | 0.00 | 0.00 | 0.00 | 0.00 |
| *Open-source LLMs* | | | | | | | | | | | | |
| Deepseek-V3.1 | 17.78 | 19.01 | 15.20 | 29.00 | 17.08 | 6.81 | 8.77 | 17.00 | 18.90 | 12.64 | 13.05 | 21.50 |
| gpt-oss-120B | 3.83 | 4.71 | 3.08 | 4.00 | 22.18 | **24.68** | **22.83** | **52.50** | 32.93 | 30.81 | 27.63 | 44.00 |
| gpt-oss-20B | 10.49 | 2.68 | 3.46 | 3.00 | 21.86 | 15.74 | 16.37 | 42.00 | 20.73 | 12.82 | 12.98 | 30.50 |
| Llama-3.1-8B | 5.60 | 9.27 | 4.58 | 10.00 | 7.86 | 10.66 | 7.83 | 25.50 | 13.39 | 12.85 | 9.88 | 19.00 |
| Qwen3-4B | 3.44 | 2.69 | 2.24 | 7.00 | 4.50 | 9.81 | 5.15 | 24.00 | 2.19 | 1.58 | 1.41 | 3.50 |
| Qwen3-4B-think | 6.82 | 1.61 | 1.94 | 3.00 | 10.35 | 7.86 | 6.89 | 22.00 | 3.77 | 1.61 | 1.54 | 3.00 |
| Qwen3-8B | 2.24 | 2.46 | 1.56 | 2.50 | 9.94 | 7.72 | 7.31 | 25.50 | 9.04 | 3.54 | 3.90 | 9.00 |
| Qwen3-8B-think | 9.22 | 7.29 | 4.22 | 7.50 | 13.30 | 14.55 | 12.08 | 31.50 | 9.66 | 12.86 | 8.98 | 20.00 |
| Qwen3-14B | 9.51 | 9.51 | 6.50 | 12.00 | 10.83 | 14.06 | 8.92 | 30.50 | 15.39 | 7.25 | 7.82 | 14.50 |
| Qwen3-14B-think | 10.33 | 6.25 | 6.15 | 7.50 | 17.97 | 13.18 | 12.23 | 37.00 | 19.42 | 12.43 | 10.60 | 25.00 |
| Qwen3-30B-A3B | 5.72 | 7.70 | 4.73 | 10.00 | 9.53 | 10.89 | 9.05 | 37.50 | 9.77 | 7.94 | 7.22 | 26.00 |
| Qwen3-30B-A3B-think | 1.44 | 1.69 | 1.17 | 3.50 | 11.48 | 14.86 | 10.70 | 33.50 | 14.21 | 17.88 | 12.77 | 33.50 |
| Qwen3-32B | 9.26 | 10.88 | 5.91 | 9.50 | 12.84 | 12.79 | 10.80 | 36.50 | 12.52 | 12.60 | 8.56 | 23.50 |
| Qwen3-32B-think | 8.54 | 11.04 | 6.39 | 10.00 | 10.21 | 15.94 | 11.04 | 40.50 | 10.77 | 14.74 | 9.80 | 32.00 |

Table 12: Results of the differential expression task (CLS & QA).

| Setting | CLS | | QA | | | | | | | | | | | |
|---|---|---|---|---|---|---|---|---|---|---|---|---|---|---|
| Model | CLS | | Overall | | | | Pos | | | | Neg | | | |
| | Rank | Acc | Recall | Prec | F1 | Acc | Recall | Prec | F1 | Acc | Recall | Prec | F1 | Acc |
| *Closed-source LLMs* | | | | | | | | | | | | | | |
| GPT-5 | 1.98 | **49.20** | **65.30** | 65.30 | 72.82 | **62.19** | 36.60 | 85.92 | 85.92 | 51.33 | 94.00 | 59.72 | 59.72 | 73.04 |
| Gemini-2.5-pro | 2.07 | 44.20 | 58.20 | 58.20 | 67.81 | 51.89 | **94.60** | 54.87 | 54.87 | **69.46** | 21.80 | **80.74** | **80.74** | 34.33 |
| Gemini-2.5-flash | 2.16 | 34.60 | 55.40 | 55.40 | 63.21 | 51.72 | 86.20 | 54.49 | 54.49 | 66.77 | 24.60 | 71.93 | 71.93 | 36.66 |
| *Domain-specific LLMs* | | | | | | | | | | | | | | |
| cell-o1 | 2.71 | 21.00 | 49.30 | 49.30 | 46.85 | 37.06 | 93.40 | 49.63 | 49.63 | 64.82 | 5.20 | 44.07 | 44.07 | 9.30 |
| rBio1 | 2.52 | 28.00 | 51.60 | 51.60 | 52.45 | 47.02 | 22.20 | 53.88 | 53.88 | 31.44 | 81.00 | 51.01 | 51.01 | 62.60 |
| C2S | - | 0.00 | 0.00 | 0.00 | 0.00 | 0.00 | 0.00 | 0.00 | 0.00 | 0.00 | 0.00 | 0.00 | 0.00 | 0.00 |
| *Open-source LLMs* | | | | | | | | | | | | | | |
| Deepseek-V3.1 | 2.14 | 11.40 | 20.50 | 20.50 | 73.29 | 29.25 | 9.20 | 85.19 | 85.19 | 16.61 | 31.80 | 61.39 | 61.39 | 41.90 |
| gpt-oss-120B | **1.95** | 47.20 | 58.70 | 58.70 | **75.47** | 51.13 | 19.00 | **95.96** | **95.96** | 31.72 | **98.40** | 54.97 | 54.97 | 70.54 |
| gpt-oss-20B | 1.99 | 26.60 | 51.30 | 51.30 | 70.50 | 50.00 | 21.20 | 85.48 | 85.48 | 33.97 | 81.40 | 55.53 | 55.53 | 66.02 |
| Llama-3.1-8B | 2.39 | 28.20 | 51.20 | 51.20 | 52.23 | 46.40 | 20.80 | 53.33 | 53.33 | 29.93 | 81.60 | 51.13 | 51.13 | 62.87 |
| Qwen3-4B | 2.34 | 33.40 | 56.50 | 56.50 | 58.08 | 54.33 | 34.60 | 61.57 | 61.57 | 44.30 | 78.40 | 54.60 | 54.60 | 64.37 |
| Qwen3-4B-think | 2.41 | 30.40 | 56.80 | 56.80 | 59.03 | 54.89 | 35.00 | 62.95 | 62.95 | 44.99 | 78.60 | 55.12 | 55.12 | 64.80 |
| Qwen3-8B | 2.31 | 35.40 | 58.60 | 58.60 | 58.64 | 58.56 | 55.40 | 59.19 | 59.19 | 57.23 | 61.80 | 58.08 | 58.08 | 59.88 |
| Qwen3-8B-think | 2.43 | 31.60 | 59.40 | 59.40 | 65.39 | 55.02 | 28.20 | 75.00 | 75.00 | 40.99 | 90.60 | 55.79 | 55.79 | 69.05 |
| Qwen3-30B-A3B | 2.30 | 36.60 | 55.40 | 55.40 | 60.83 | 49.01 | 20.00 | 68.49 | 68.49 | 30.96 | 90.80 | 53.16 | 53.16 | 67.06 |
| Qwen3-30B-A3B-think | 2.32 | 34.80 | 56.00 | 56.00 | 70.38 | 46.58 | 14.00 | 87.50 | 87.50 | 24.14 | 98.00 | 53.26 | 53.26 | **98.00** |
| Qwen3-32B | 2.33 | 34.40 | 57.30 | 57.30 | 66.92 | 50.23 | 19.60 | 79.67 | 79.67 | 31.46 | 95.00 | 54.16 | 54.16 | 68.99 |
| Qwen3-32B-think | 2.33 | 34.80 | 55.90 | 55.90 | 65.96 | 47.65 | 16.20 | 78.64 | 78.64 | 26.87 | 95.60 | 53.29 | 53.29 | 68.43 |

Table 13: Results of the direction of change task.

| Model | Overall | | | | Increase | | | | Decrease | | | |
|---|---|---|---|---|---|---|---|---|---|---|---|---|
| | Acc | Rec | Prec | F1 | Acc | Rec | Prec | F1 | Acc | Rec | Prec | F1 |
| *Closed-source LLMs* | | | | | | | | | | | | |
| GPT-5 | 55.20 | **55.80** | 55.84 | 55.19 | 62.33 | 49.27 | 50.54 | 61.14 | 49.27 | 61.14 | **61.14** | 54.56 |
| Gemini-2.5-pro | 52.60 | 53.57 | 54.12 | 52.66 | 64.10 | 43.04 | 48.74 | 59.49 | 43.04 | 59.49 | 59.49 | 49.95 |
| Gemini-2.5-flash | 49.00 | 49.49 | 56.60 | 52.24 | 54.85 | 44.14 | 50.92 | **62.27** | 44.14 | 62.27 | 62.27 | 51.66 |
| *Domain-specific LLMs* | | | | | | | | | | | | |
| cell-o1 | 46.10 | 46.86 | 47.93 | 46.55 | 55.07 | 38.64 | 43.63 | 52.23 | 38.64 | 52.23 | 52.23 | 44.42 |
| rBio1 | 53.20 | 52.41 | 52.80 | 52.50 | 43.83 | 60.99 | 48.77 | 56.83 | 60.99 | 56.83 | 56.83 | 58.83 |
| C2S | 0.00 | 0.00 | 0.00 | 0.00 | 0.00 | 0.00 | 0.00 | 0.00 | 0.00 | 0.00 | 0.00 | 0.00 |
| *Open-source LLMs* | | | | | | | | | | | | |
| Deepseek-V3.1 | 14.20 | 14.86 | 54.17 | 21.71 | 22.03 | 7.69 | 45.66 | 62.69 | 7.69 | 62.69 | 62.69 | 13.70 |
| gpt-oss-120B | 55.40 | 55.00 | 55.57 | **55.28** | 50.66 | 59.34 | 51.69 | 59.45 | 59.34 | 59.45 | 59.45 | 59.40 |
| gpt-oss-20B | 49.00 | 48.10 | 55.82 | 51.45 | 38.33 | 57.88 | 51.79 | 59.85 | 57.88 | 59.85 | 59.85 | 58.85 |
| Llama-3.1-8B | 39.70 | 37.67 | 47.81 | 39.59 | 15.64 | 59.71 | 42.26 | 53.36 | 59.71 | 53.36 | 53.36 | 56.35 |
| Qwen3-4B | 52.70 | 50.75 | 52.45 | 49.87 | 29.52 | 71.98 | 49.08 | 55.82 | 71.98 | 55.82 | 55.82 | 62.88 |
| Qwen3-4B-think | 53.10 | 52.75 | 55.06 | 53.88 | 48.90 | 56.59 | 50.92 | 59.20 | 56.59 | 59.20 | 59.20 | 57.87 |
| Qwen3-8B | 53.30 | 51.93 | 53.02 | 51.78 | 37.00 | 66.85 | 49.70 | 56.33 | 66.85 | 56.33 | 56.33 | 61.14 |
| Qwen3-8B-think | 55.20 | 54.89 | 54.88 | 54.88 | 51.54 | 58.24 | 50.65 | 59.11 | 58.24 | 59.11 | 59.11 | 58.67 |
| Qwen3-30B-A3B | **56.40** | 53.80 | **57.58** | 51.57 | 25.55 | **82.05** | **57.43** | 57.73 | 82.05 | 57.73 | 57.73 | **67.78** |
| Qwen3-30B-A3B-think | 53.40 | 52.04 | 52.22 | 51.54 | 37.22 | 66.85 | 48.29 | 56.15 | 66.85 | 56.15 | 56.15 | 61.04 |
| Qwen3-32B | 51.80 | 51.72 | 51.96 | 51.78 | 50.88 | 52.56 | 47.53 | 56.39 | 52.56 | 56.39 | 56.39 | 54.41 |
| Qwen3-32B-think | 53.30 | **54.27** | 54.41 | 53.16 | **64.76** | 43.77 | 48.92 | 59.90 | **64.76** | 59.90 | 59.90 | 50.58 |

Table 14: Results of the perturbation prediction with HVGs task.

| | CLS | | | | | QA | | | | | | |
|---|---|---|---|---|---|---|---|---|---|---|---|---|
| Model | Acc | Acc | Recall | Prec | F1 | No Rec | Yes Rec | No Prec | Yes Prec | No F1 | Yes F1 |
| *Closed-source LLMs* | | | | | | | | | | | |
| GPT-5 | 47.33 | **53.67** | 53.67 | 54.80 | **50.75** | 78.00 | 29.33 | 52.47 | 57.14 | 62.73 | 38.77 |
| Gemini-2.5-pro | 23.33 | 39.33 | 39.33 | 60.62 | 45.17 | 24.00 | 54.67 | **65.45** | 55.78 | 35.12 | 55.22 |
| Gemini-2.5-flash | 7.33 | 11.67 | 11.67 | 55.74 | 19.02 | 8.67 | 14.67 | 59.09 | 52.38 | 15.12 | 22.92 |
| *Domain-specific LLMs* | | | | | | | | | | | |
| cell-o1 | 47.33 | 51.33 | 51.33 | 52.41 | 47.56 | 24.00 | **78.67** | 53.73 | 51.08 | 33.18 | **61.94** |
| rBio1 | 52.67 | 46.33 | 46.33 | 45.66 | 44.17 | 66.00 | 26.67 | 47.37 | 43.96 | 55.15 | 33.20 |
| C2S | 0.00 | 0.00 | 0.00 | 0.00 | 0.00 | 0.00 | 0.00 | 0.00 | 0.00 | 0.00 | 0.00 |
| *Open-source LLMs* | | | | | | | | | | | |
| Deepseek-V3.1 | 18.0 | 17.67 | 17.67 | 35.71 | 22.21 | 34.67 | 0.67 | 57.14 | 14.29 | 43.15 | 1.27 |
| gpt-oss-120B | 47.33 | 49.0 | 49.0 | 59.75 | 39.16 | 90.67 | 7.33 | 50.75 | **68.75** | 65.07 | 13.25 |
| gpt-oss-20B | 8.67 | 23.33 | 23.33 | 57.02 | 26.39 | 44.67 | 2.00 | 54.03 | 60.00 | 48.91 | 3.87 |
| Llama-3.1-8B | 50.67 | 48.00 | 48.00 | 42.12 | 35.54 | 92.00 | 4.00 | 48.94 | 35.29 | 63.89 | 7.19 |
| Qwen3-4B | 39.33 | 48.33 | 48.33 | 44.73 | 36.79 | 91.33 | 5.33 | 49.46 | 40.0 | 64.17 | 9.41 |
| Qwen3-4B-think | 7.33 | 29.00 | 29.00 | 39.74 | 28.64 | 54.67 | 3.33 | 48.24 | 31.25 | 51.25 | 6.02 |
| Qwen3-8B | 49.33 | 50.0 | 50.0 | 50.54 | 44.96 | 80.67 | 19.33 | 50.21 | 50.88 | 61.89 | 28.02 |
| Qwen3-8B-think | 48.00 | 51.33 | 51.33 | 57.49 | 38.74 | 96.67 | 6.00 | 50.70 | 64.29 | **66.51** | 10.98 |
| Qwen3-30B-A3B | 46.67 | 48.00 | 48.00 | 39.65 | 34.21 | 94.00 | 2.00 | 49.30 | 30.00 | 64.68 | 3.75 |
| Qwen3-30B-A3B-think | 48.67 | 49.67 | 49.67 | 24.92 | 33.18 | **99.33** | 0.00 | 49.83 | 0.00 | 66.37 | 0.00 |
| Qwen3-32B | 50.67 | 49.67 | 49.67 | 48.91 | 38.75 | 92.00 | 7.33 | 50.00 | 47.83 | 64.79 | 12.72 |
| Qwen3-32B-think | **54.67** | 50.0 | 50.0 | 50.0 | 38.87 | 92.67 | 7.33 | 50.0 | 50.0 | 64.95 | 12.79 |

Table 15: Results of the perturbed cell generation task.

| Model | Cosine Sim. | Top@50 | Top@100 | nDCG@50 | nDCG@100 | Spearman@50 | Spearman@100 |
|---|---|---|---|---|---|---|---|
| *Closed-source LLMs* | | | | | | | |
| GPT-5 | 99.00 | 77.57 | 83.02 | 96.55 | 96.90 | 72.62 | 76.47 |
| Gemini-2.5-pro | 95.85 | 42.23 | 51.08 | 53.72 | 57.62 | 63.18 | 70.95 |
| Gemini-2.5-flash | 81.03 | 6.44 | 19.69 | 24.24 | 12.69 | 11.35 | 13.23 |
| *Domain-specific LLMs* | | | | | | | |
| cell-o1 | 74.31 | 23.89 | 18.55 | 54.12 | 37.35 | 62.48 | 62.22 |
| rBio1 | 77.66 | 39.25 | 35.17 | 53.58 | 46.86 | 66.47 | 65.60 |
| C2S | - | 0.00 | 0.00 | 0.00 | 0.00 | 0.00 | 0.00 |
| *Open-source LLMs* | | | | | | | |
| Deepseek-V3.1 | **99.23** | 85.88 | **89.44** | 98.41 | **98.75** | 86.66 | 89.56 |
| gpt-oss-120B | 94.45 | 6.39 | 20.29 | 9.77 | 18.64 | 24.68 | 27.89 |
| gpt-oss-20B | 78.87 | 5.48 | 10.91 | 6.54 | 10.15 | 47.11 | 34.78 |
| Llama-3.1-8B | 82.76 | 26.20 | 33.68 | 45.45 | 46.15 | 43.02 | 52.17 |
| Qwen3-4B | 90.66 | 70.43 | 71.55 | 80.59 | 78.94 | 83.36 | 83.84 |
| Qwen3-4B-think | 68.65 | 22.25 | 20.57 | 38.55 | 29.91 | 50.49 | 50.46 |
| Qwen3-8B | 89.04 | 34.20 | 37.75 | 58.55 | 51.80 | 50.80 | 41.84 |
| Qwen3-8B-think | 76.36 | 18.52 | 17.97 | 47.00 | 34.90 | 33.60 | 37.40 |
| Qwen3-30B-A3B | 91.09 | 53.44 | 53.02 | 59.25 | 58.70 | 73.01 | 72.06 |
| Qwen3-30B-A3B-think | 74.84 | 27.35 | 25.59 | 47.52 | 37.76 | 35.05 | 43.03 |
| Qwen3-32B | 97.45 | 72.21 | 72.61 | 90.39 | 85.71 | 81.08 | 79.18 |
| Qwen3-32B-think | 97.73 | **86.07** | 87.27 | **99.02** | 96.94 | **88.85** | **89.86** |

Table 16: Results of the drug response task.

| Setting | Overall | | | | Sensitive | | | Resistant | | |
|---|---|---|---|---|---|---|---|---|---|---|
| Metric | Precision | Recall | F1 | Accuracy | Precision | Recall | F1 | Precision | Recall | F1 |
| *Closed-source LLMs* | | | | | | | | | | |
| GPT-5 | 24.68 | 48.75 | 32.77 | 48.75 | 0.00 | **97.50** | 0.00 | 49.37 | 0.00 | 65.55 |
| Gemini-2.5-pro | 41.51 | 49.38 | 34.12 | 49.38 | 1.25 | **97.50** | 33.33 | 49.68 | 2.41 | **65.82** |
| Gemini-2.5-flash | 40.30 | 16.88 | 21.82 | 16.88 | 5.00 | 28.75 | 36.36 | 44.23 | 8.79 | 34.85 |
| *Domain-specific LLMs* | | | | | | | | | | |
| cell-o1 | 47.58 | 30.63 | 36.49 | 30.63 | 18.75 | 42.50 | 42.86 | 52.31 | 26.09 | 46.90 |
| rBio1 | 48.48 | 46.88 | 44.17 | 46.88 | **71.25** | 22.50 | 48.31 | 48.65 | **57.58** | 30.77 |
| C2S | 0.00 | 0.00 | 0.00 | 0.00 | 0.00 | 0.00 | 0.00 | 0.00 | 0.00 | 0.00 |
| *Open-source LLMs* | | | | | | | | | | |
| Deepseek-V3.1 | 50.00 | **50.00** | 39.75 | 50.00 | 8.75 | 91.25 | 50.00 | 50.00 | 14.89 | 64.60 |
| gpt-oss-120B | **55.16** | 50.62 | 36.72 | **50.62** | 3.75 | **97.50** | 60.00 | 50.32 | 7.06 | 66.38 |
| gpt-oss-20B | 52.01 | **50.00** | 41.70 | 50.00 | 11.25 | 88.75 | 52.94 | 51.08 | 18.56 | 64.84 |
| Llama-3.1-8B | 53.57 | 26.88 | 34.96 | 26.88 | 20.00 | 33.75 | 57.14 | 50.00 | 29.63 | 40.30 |
| Qwen3-4B | 47.26 | 41.88 | 37.66 | 41.88 | 11.25 | 72.50 | 47.37 | 47.15 | 18.18 | 57.14 |
| Qwen3-4B-think | 41.01 | 43.75 | 36.44 | 43.75 | 8.75 | 78.75 | 35.00 | 47.01 | 14.00 | 58.88 |
| Qwen3-8B-think | 44.67 | 48.75 | 36.62 | 48.75 | 5.00 | 92.50 | 40.00 | 49.33 | 8.89 | 64.35 |
| Qwen3-14B | 42.06 | 43.75 | 39.83 | 43.75 | 17.50 | 70.00 | 37.84 | 46.28 | 23.93 | 55.72 |
| Qwen3-14B-think | 43.46 | 46.25 | 38.83 | 46.25 | 11.25 | 81.25 | 39.13 | 47.79 | 17.48 | 60.19 |
| Qwen3-30B-A3B | 43.63 | 16.88 | 24.32 | 16.88 | 16.25 | 17.50 | 44.83 | 42.42 | 23.85 | 24.78 |
| Qwen3-30B-A3B-think | 38.26 | 46.25 | 35.23 | 46.25 | 5.00 | 87.50 | 28.57 | 47.95 | 8.51 | 61.95 |
| Qwen3-32B | 50.82 | 45.00 | **46.24** | 45.00 | 31.25 | 58.75 | 53.19 | 48.45 | **39.37** | 53.11 |
| Qwen3-32B-think | 35.27 | 46.25 | 33.69 | 46.25 | 2.50 | 90.00 | 22.22 | 48.32 | 4.49 | 62.88 |

Table 17: Results of the conditional cell generation task.

| Model | Cosine Sim. | Top@50 | Top@100 | nDCG@50 | nDCG@100 | Spearman@50 | Spearman@100 |
|---|---|---|---|---|---|---|---|
| *Closed-source LLMs* | | | | | | | |
| GPT-5 | 93.56 | 24.21 | 33.82 | **97.74** | **94.70** | 25.93 | 13.18 |
| Gemini-2.5-pro | **93.75** | **36.47** | **40.32** | 86.30 | 89.01 | 41.61 | 34.72 |
| Gemini-2.5-flash | 74.70 | 7.37 | 9.50 | 32.23 | 31.72 | 11.33 | 12.70 |
| *Domain-specific LLMs* | | | | | | | |
| cell-o1 | 60.99 | 4.37 | 3.47 | 47.83 | 32.10 | -5.67 | 26.30 |
| rBio1 | 63.47 | 0.53 | 0.66 | 52.81 | 35.38 | 50.00 | 23.33 |
| C2S | - | 0.00 | 0.00 | 0.00 | 0.00 | 0.00 | 0.00 |
| *Open-source LLMs* | | | | | | | |
| Deepseek-V3.1 | 89.80 | 11.63 | 17.00 | 62.25 | 68.36 | 6.56 | 18.75 |
| gpt-oss-120B | 91.18 | 14.58 | 22.45 | 65.60 | 66.85 | 8.03 | 18.77 |
| gpt-oss-20B | 84.03 | 5.58 | 10.71 | 37.47 | 36.70 | 17.62 | 15.65 |
| Llama-3.1-8B | 60.86 | 2.11 | 1.39 | 55.08 | 34.81 | 17.43 | 31.19 |
| Qwen3-4B | 68.13 | 0.95 | 0.68 | 44.01 | 28.95 | 15.00 | **52.50** |
| Qwen3-4B-think | 65.75 | 1.32 | 1.32 | 41.03 | 29.15 | **55.43** | 32.08 |
| Qwen3-8B | 80.50 | 2.89 | 2.82 | 70.94 | 61.18 | 6.73 | 30.85 |
| Qwen3-8B-think | 71.68 | 3.37 | 2.84 | 70.68 | 49.63 | -17.55 | -2.06 |
| Qwen3-30B-A3B | 75.62 | 4.11 | 4.68 | 75.96 | 57.56 | 34.70 | 29.42 |
| Qwen3-30B-A3B-think | 77.67 | 2.95 | 2.71 | 75.18 | 54.93 | -4.71 | 6.62 |
| Qwen3-32B | 89.78 | 6.37 | 8.24 | 84.37 | 79.37 | 3.64 | 19.58 |
| Qwen3-32B-think | 80.91 | 7.00 | 8.34 | 84.17 | 63.65 | -5.21 | 6.11 |

Table 18: Results of the gene annotation task.

| Model | BERTScore P | BERTScore R | BERTScore F1 | ROUGE-1 R |
|---|---|---|---|---|
| *Closed-source LLMs* | | | | |
| GPT-5 | 67.19 | 68.98 | 68.00 | 43.48 |
| Gemini-2.5-pro | 67.61 | 68.87 | 68.15 | 58.70 |
| Gemini-2.5-flash | 68.10 | 69.18 | 68.56 | 41.30 |
| *Domain-specific LLMs* | | | | |
| cell-o1 | 64.47 | 68.00 | 65.95 | 63.04 |
| rBio1 | **68.91** | 69.62 | **69.17** | 30.43 |
| C2S | 68.86 | 59.47 | 63.74 | 19.57 |
| *Open-source LLMs* | | | | |
| Deepseek-V3.1 | 68.00 | **69.86** | 68.83 | 41.30 |
| gpt-oss-120B | 65.34 | 68.41 | 66.73 | 71.74 |
| gpt-oss-20B | 61.74 | 66.28 | 63.82 | **95.65** |
| Llama-3.1-8B | 65.94 | 69.03 | 67.37 | 63.04 |
| Qwen3-4B | 58.97 | 67.01 | 62.66 | 41.30 |
| Qwen3-4B-think | 66.80 | 68.80 | 67.70 | 52.17 |
| Qwen3-8B | 68.67 | 69.84 | 69.18 | 52.17 |
| Qwen3-8B-think | 67.75 | 68.90 | 68.25 | 43.48 |
| Qwen3-30B-A3B | 67.97 | 68.92 | 68.35 | 43.48 |
| Qwen3-30B-A3B-think | 67.82 | 69.17 | 68.40 | 34.78 |
| Qwen3-32B | 67.78 | 68.90 | 68.25 | 56.52 |
| Qwen3-32B-think | 67.35 | 69.71 | 68.43 | 39.13 |

Table 19: Results of the few shot ablation task.

| Model | leak | | | | mask | | | |
|---|---|---|---|---|---|---|---|---|
| | Accuracy | Precision | Recall | F1 | Accuracy | Precision | Recall | F1 |
| gpt-5-chat-latest | **72.00** | **69.26** | **78.74** | **71.65** | 63.00 | 60.51 | **72.58** | **61.36** |
| cell_o1 | 51.00 | 52.18 | 54.44 | 48.97 | **54.50** | **55.33** | 61.09 | **53.23** |
| qwen3-32b | 48.50 | 46.59 | 56.09 | 46.61 | 37.00 | 35.10 | 42.89 | 33.02 |
| qwen3-8b | 36.50 | 35.15 | 47.99 | 32.23 | 31.00 | 29.62 | 40.94 | 26.93 |

Table 20: Results of the robustness ablation task.

| Model | 0% | | | | 20% | | | | 40% | | | |
|---|---|---|---|---|---|---|---|---|---|---|---|---|
| | Accuracy | Precision | Recall | F1 | Accuracy | Precision | Recall | F1 | Accuracy | Precision | Recall | F1 |
| gpt-5-chat-latest | **67.50** | **65.17** | **81.06** | **67.10** | **68.00** | **66.39** | **81.12** | **67.96** | **68.50** | **66.59** | **81.43** | **68.45** |
| cell_o1 | 47.00 | 47.90 | 54.31 | 44.00 | 45.00 | 46.04 | 56.56 | 43.03 | 43.00 | 43.76 | 53.63 | 40.89 |
| qwen3-32b | 45.00 | 43.96 | 51.70 | 42.45 | 44.00 | 43.47 | 50.77 | 41.44 | 48.00 | 47.37 | 58.63 | 46.23 |
| qwen3-8b | 29.50 | 28.15 | 37.11 | 23.45 | 35.00 | 34.17 | 41.90 | 29.97 | 34.50 | 33.20 | 40.56 | 29.91 |

| Model | 60% | | | | 80% | | | | 100% | | | |
|---|---|---|---|---|---|---|---|---|---|---|---|---|
| | Accuracy | Precision | Recall | F1 | Accuracy | Precision | Recall | F1 | Accuracy | Precision | Recall | F1 |
| gpt-5-chat-latest | **67.50** | **65.77** | 79.62 | **68.53** | **68.50** | **66.53** | 78.02 | **68.85** | **70.00** | **68.15** | **80.36** | **70.52** |
| cell_o1 | 47.00 | 48.24 | 53.14 | 45.38 | 44.50 | 44.61 | **55.94** | 42.40 | 46.50 | 47.35 | 47.83 | 42.75 |
| qwen3-32b | 44.00 | 42.78 | **56.99** | 43.00 | 41.00 | 39.99 | 50.25 | 37.84 | 43.50 | 42.94 | 51.91 | 41.92 |
| qwen3-8b | 33.50 | 32.48 | 38.05 | 27.50 | 30.50 | 29.28 | 32.75 | 24.54 | 31.00 | 29.37 | 35.07 | 25.99 |

Table 21: Results of the gene list length ablation task.

| Model | 20 | | | | 50 | | | | 100 | | | | 200 | | | | 500 | | | |
|---|---|---|---|---|---|---|---|---|---|---|---|---|---|---|---|---|---|---|---|---|
| | Acc | Prec | Rec | F1 | Acc | Prec | Rec | F1 | Acc | Prec | Rec | F1 | Acc | Prec | Rec | F1 | Acc | Prec | Rec | F1 |
| gpt-5-chat-latest | **63.50** | **63.10** | **75.36** | **63.14** | **68.50** | **66.77** | **80.64** | **67.79** | **71.00** | **68.19** | **80.61** | **69.04** | **75.00** | **71.42** | **84.75** | **72.32** | **71.50** | **67.81** | **80.76** | **68.89** |
| cell_o1 | 41.00 | 43.20 | 52.29 | 39.19 | 46.50 | 47.48 | 57.17 | 45.05 | 47.50 | 47.97 | 47.83 | 42.87 | 46.00 | 46.75 | 52.81 | 42.42 | 42.00 | 42.25 | 47.26 | 38.66 |
| qwen3-32b | 48.00 | 47.41 | 57.79 | 46.05 | 46.50 | 45.65 | 52.63 | 43.85 | 46.50 | 45.92 | 54.91 | 44.62 | 47.00 | 46.47 | 60.51 | 45.86 | 46.00 | 44.73 | 55.51 | 43.40 |
| qwen3-8b | 33.50 | 33.47 | 37.34 | 29.47 | 34.00 | 33.38 | 40.54 | 28.06 | 30.00 | 28.85 | 38.16 | 25.20 | 34.50 | 33.46 | 43.74 | 31.13 | 30.00 | 28.37 | 37.19 | 24.87 |

## A.5 PROMPT TEMPLATES

---

**Cell Type Annotation**

**System Prompt:**
You are a helpful biology expert. You now need to determine the most probable cell type of a single cell based on its expression profile, which is represented by the top {n_genes} highly expressed genes sorted in descending order.

**Task:**

- **Goal:** Given the expression profile of a single cell, identify its most probable cell type.
- **Context:**
    1. A list of the top {n_genes} genes with the highest expression in one single cell. These highly expressed genes serve as a proxy for cell identity and can be used to infer the most likely cell type.
    2. The gene list is obtained from **single-cell RNA sequencing**, with genes sorted in descending order of expression.

**Answer Choices:** {cell_type_list}

**Guidance:**

- Use biological knowledge to reason about what cell type the given expression profile most likely represents.
- The provided gene list reflects the single-cell expression profile, which serves as a reference for inferring biological pathways and regulatory interactions.
- Decide independently which perspective (e.g., marker genes, pathways, expression specificity) to use for reasoning.

**Output Rules:**

- Put your reasoning process inside `<analyze></analyze>` tags.
- Put your final decision inside `<answer></answer>` tags.
- The `<answer>` tag must contain exactly one option from the provided list.

**User Prompt:**
From single-cell RNA sequencing, the expression profile of this cell was ranked in descending order, and the top {n_genes} highly expressed genes were selected:
{cell_sentence}

Please tell me what the cell type is.

---

**Batch Cell Type Annotation Task**

**System Prompt:**

You are an expert assistant specialized in cell type annotation. You will be given a batch of N cells from the same donor, where each cell represents a unique cell type. For each cell, the top expressed genes are provided in descending order of expression. Using both the gene expression data and donor information, determine the correct cell type for each cell. You will also receive a list of N candidate cell types, and each candidate must be assigned to exactly one cell. Ensure that you consider all cells and candidate types together, rather than annotating each cell individually.

Include your detailed reasoning within `<think></think>` tags, and provide your final answer within `<answer></answer>` tags. The final answer should be a single string listing the assigned cell types in order, separated by "—".

**User Context:**

The cell is from a female at the 73-year-old stage, originating from the lung. The patient has been diagnosed with chronic obstructive pulmonary disease. The patient is a smoker. There is no cancer present.

**Cells:**

- Cell 1: ...
- Cell 2: ...
- Cell 3: ...
- Cell 4: ...
- Cell 5: ...
- Cell 6: ...
- Cell 7: ...
- Cell 8: ...
- Cell 9: ...

**Candidate Cell Types:**

- classical monocyte
- endothelial cell of lymphatic vessel
- epithelial cell of lung
- fibroblast of lung
- macrophage
- malignant cell
- non-classical monocyte
- pulmonary artery endothelial cell
- smooth muscle cell

**Multi-Omics Cell Type Annotation**

**System Prompt:**

You are a helpful biology expert.

**Task**

- **Goal**: Predict the most probable cell type for a single cell using integrated transcriptomic and proteomic profiles from multi-omics sequencing.
- **Context**: You will be provided with the following information:
    1. A transcriptomic profile containing the top most highly expressed protein-coding genes (non-zero expression), sorted in descending order by expression level.
    2. A proteomic profile listing all non-zero surface proteins detected via CITE-seq (134 possible targets), sorted in descending order by protein expression level.
    3. The complete list of candidate cell types.

**Guidance**

Use your biological knowledge to reason about the relationships between transcriptomic and proteomic signals, including marker genes, protein markers, and cell-type-specific pathways.

**Output Rules**

- Put your reasoning process inside `<analyze></analyze>` tags.
- Put your final answer inside `<answer></answer>` tags.
- The `<answer>` tag must contain exactly one valid cell type selected from the provided candidate list.

**User Prompt:**

Task: Single-Cell Type Annotation Using Multi-Omics Data.

You are given paired transcriptomic and proteomic profiles from a single cell, generated using multi-omics single-cell sequencing (including CITE-seq for surface protein detection).

- The transcriptomic data includes the top 200 most highly expressed protein-coding genes (non-zero expression), sorted in descending order by expression level.
- The proteomic data includes all 134 measurable surface proteins via CITE-seq, with only non-zero detections provided, sorted in descending order by protein expression level.

Possible cell types: {cell_type_list}

Transcriptomic profile (top 200 genes): {gene_top_names}

Proteomic profile (non-zero surface proteins, ranked by expression): {protein_top_names}

Based on this combined multi-omics signature, please predict the most likely cell type.

**Perturbation QA**

**System Prompt:**

You are a helpful biology expert. You are given a task of assessing whether CRISPRi knockdown of a perturbation gene leads to a statistically significant change in the expression of another gene.

**Task**

- **Goal**: Determine if the queried target gene shows a significant expression change after perturbing the specified gene using CRISPRi.
- **Context**: You will be provided with the following information:
    1. The **background cell line** where the perturbation is applied.
    2. A list of **highly expressed genes** in this cell line, obtained by averaging single-cell expression profiles across all cells (a bulk-like profile representing the baseline transcriptional state). The list is sorted in descending order of mean expression.
    3. The **perturbation condition** (which gene is knocked down by CRISPRi).
    4. The **target gene** whose potential expression change should be assessed.

**Guidance**

- Use your biological knowledge to reason about the potential relationship between the perturbation gene and the target gene.
- The provided gene list reflects the average expression profile of the {cell_line}, serving as a reference for inferring biological pathways and regulatory interactions.
- Knocking down a gene may disrupt its associated pathways, which can in turn affect the expression of the target gene.
- While most genes typically remain unchanged under a given perturbation, in this testing setup the probability has been artificially adjusted—around **50%** of the queries involve target genes that do show significant changes.
- Based on your biological knowledge, independently determine from which perspective to conduct the analysis.

**Output Rules**

- Put your reasoning process inside `<analyze></analyze>` tags.
- Put your final decision inside `<answer></answer>` tags.
- The `<answer>` tag must contain one of the following:
    - `<answer>Yes</answer>`: if the target gene shows a statistically significant change.
    - `<answer>No</answer>`: if the target gene does not show a statistically significant change.

**User Prompt:**
In {cell_line} cells, we apply CRISPRi knockdown of {perturbation}.
Here is the list of highly expressed genes in this background, sorted by mean expression (high → low):
{cell_sentence}

Does {gene} show a significant change in expression under this perturbation?

**Perturbation CLS**

**System Prompt:**

You are a helpful biology expert. You are given a task of assessing whether CRISPRi knockdown of a perturbation gene leads to a statistically significant change in the expression of another gene.

**Task**

- **Goal**: From the provided candidate genes, select the **single most likely gene** that shows a statistically significant expression change after CRISPRi perturbation of the specified gene.
- **Context**: You will be provided with:
    1. The **background cell line** where the perturbation is applied.
    2. A list of **highly expressed genes** in this cell line, obtained by averaging single-cell expression profiles across all cells (bulk-like baseline). The list is sorted in descending order of mean expression.
    3. The **perturbation condition** (which gene is knocked down by CRISPRi).
    4. A **candidate set of gene names**: {candidates}.

**Guidance**

- Use biological knowledge to reason about mechanistic links (e.g., pathways, transcription factor–target relationships, co-expression, compensatory mechanisms, or cell-type specificity).
- The background gene list reflects the average expression profile of {cell_line}, which can guide inference of pathway or regulatory interactions.
- In this testing setup, the candidate set follows a fixed rule:
    - **One gene** is strongly differentially expressed (FDR < 0.05, Wilcoxon test with BH correction).
    - **Two genes** are weakly or borderline differentially expressed ($0.05 \leq$ FDR < 0.1).
    - **Two genes** are not differentially expressed (FDR > 0.5).
- Your task is to identify the **single gene** that is most likely the truly significant DEG.
- You must choose **exactly one** gene from {candidates}.

**Output Rules**

- Put your reasoning inside `<analyze></analyze>`. Keep it concise but biologically grounded.
- Put your final decision inside `<answer></answer>`.
- The `<answer>` tag must contain **only one exact gene symbol** selected from {candidates}.

**User Prompt:**
In {cell_line} cells, we apply CRISPRi knockdown of {perturbation}.
Here is the background high-expression gene list (high → low):
{cell_sentence}

From the candidates {candidates}, which gene most likely shows a significant expression change?

**Perturbation Direction Prediction**

**System Prompt:**
You are a helpful biology expert. You are given a task of determining the direction of expression change caused by a CRISPRi perturbation.

**Task**

- **Goal**: Decide whether a queried gene shows an **increase (Increase)** or **decrease (Decrease)** in expression after knockdown of a target gene by CRISPRi.
- **Context**: You will be provided with the following information:
    1. The **background cell line** where the perturbation is applied.
    2. A list of **highly expressed genes** in this cell line, obtained by averaging single-cell expression profiles across all cells (a bulk-like profile representing the baseline transcriptional state). The list is sorted in descending order of mean expression (genes appearing later have lower expression).
    3. The **perturbation condition** (which gene is knocked down by CRISPRi).
    4. The **target gene** (already determined to be differentially expressed) whose direction of change should be assessed.

**Guidance**

- Use your biological knowledge to reason about the potential relationship between the perturbation gene and the target gene.
- The provided gene list reflects the average expression profile of the {cell_line}, serving as a reference for inferring biological pathways and regulatory interactions.
- Knocking down a gene may disrupt its associated pathways, which can in turn affect the expression of the target gene.
- The target gene has already been identified as significantly differentially expressed based on the log2 fold change from the Wilcoxon BH test. Your task is to further determine whether it is significantly increased or decreased.
- Based on your biological knowledge, independently determine from which perspective to conduct the analysis.

**Output Rules**

- Put your reasoning process inside `<analyze></analyze>` tags.
- Put your final decision inside `<answer></answer>` tags.
- The `<answer>` tag must contain one of the following:
    - `<answer>Increase</answer>`: if the target gene will be significantly increased after CRISPRi perturbation.
    - `<answer>Decrease</answer>`: if the target gene will be significantly decreased after CRISPRi perturbation.

**User Prompt:**
In {cell_line} cells, we apply CRISPRi knockdown of {perturbation}.
Here is the list of highly expressed genes in this background, sorted by mean expression (high → low):
{cell_sentence}

Does {gene} go Increase or Decrease in expression under this perturbation?

---

**Perturbation CLS with HVGs**

**System Prompt:**
You are a helpful biology expert. You are given a task of identifying which of two candidate genes is the true perturbation that caused the observed expression change.

**Task**

- **Goal**: Choose the correct perturbation gene between the two given candidates.
- **Context**: You will be provided with:
    1. The background cell line.
    2. A **control gene list** (200 genes ranked by mean expression).
    3. A **perturbed gene list** (200 genes ranked by mean expression after perturbation).
    4. Two candidate perturbation genes: {choice_1} and {choice_2}.

**Guidance**

- Each gene list is constructed by HVG analysis followed by selection of the **top 200 bulk-expressed genes**.
- These lists capture the most informative signatures of control and perturbation states.
- Compare the control and perturbed lists carefully.
- Use biological reasoning: pathway involvement, regulatory modules, upstream regulators, compensatory shifts.
- Only one candidate is correct.

**Output Rules**

- Put your reasoning process inside `<analyze></analyze>` tags.
- Put your final decision inside `<answer></answer>` tags.
- The `<answer>` must be exactly one of the two candidates: `<answer>{choice_1}</answer>` or `<answer>{choice_2}</answer>`.

**User Prompt:**
Background: {cell_line}.
Control gene list:
{ctrl_gene_list}

Perturbed gene list:
{pert_gene_list}

Which perturbation gene is correct, {choice_1} or {choice_2}?

---

**Perturbation QA with HVGs**

**System Prompt:**
You are a helpful biology expert. You are given a task of determining whether the observed difference between the control and perturbed gene expression profiles is due to CRISPRi knockdown of a specific gene.

**Task**

- **Goal**: Judge whether the queried perturbation gene is the true cause of the expression profile change.
- **Context**: You will be provided with:
  1. The background cell line.
  2. A **control gene list** (200 genes ranked by mean expression).
  3. A **perturbed gene list** (200 genes ranked by mean expression after perturbation).
  4. A queried **perturbation gene**.

**Guidance**

- Each gene list is constructed by first performing highly variable gene (HVG) analysis, then selecting the **top 200 genes by bulk mean expression**.
- These gene lists represent the most biologically informative subsets of the expression state.
- Use your biological knowledge: consider pathways, regulatory networks, transcription factors, co-expression, compensatory responses, etc.
- Decide whether the observed differences are consistent with knockdown of the queried gene.
- About half of the queries are true perturbations (Yes), half are false (No).

**Output Rules**

- Put your reasoning process inside `<analyze></analyze>` tags.
- Put your final decision inside `<answer></answer>` tags.
- The `<answer>` must be either `<answer>Yes</answer>` or `<answer>No</answer>`.

**User Prompt:**
Background: {cell_line}.
Control gene list:
{ctrl_gene_list}

Perturbed gene list:
{pert_gene_list}

Was this caused by knockdown of {query}?

---

**Perturbed Cell Generation**

**System Prompt:**
You are a helpful biology expert. You are given a task of predicting how the gene expression profile will change under a perturbation.

**Task**

- **Goal**: Given the background cell line, the control gene list, and a perturbation gene, predict the perturbed gene list.
- **Context**: You will be provided with:
    1. The background cell line.
    2. A **control gene list** (200 genes ranked by mean expression).
    3. A **perturbation gene** to be knocked down by CRISPRi.

**Guidance**

- Each gene list is obtained by HVG analysis, then the **top 200 genes by bulk mean expression** are selected and ranked.
- These gene lists are informative signatures of cell states.
- Use your biological knowledge: pathways, transcriptional regulation, compensatory activation, stress responses, metabolic rewiring, etc.
- Always reason explicitly before generating the perturbed gene list.
- The perturbed gene list should still contain 200 genes.

**Output Rules**

- Put your reasoning process inside `<analyze></analyze>` tags.
- Put your final output inside `<answer></answer>` tags.
- The `<answer>` must be a numbered list of exactly 200 entries, one gene per line, like:

```
<answer>
1. GENE_A
2. GENE_B
...
200. GENE_Z
</answer>
```

**User Prompt:**
Background: {cell_line}.
Control gene list:
{ctrl_gene_list}

Perturbation: knockdown of {query}.

Predict the perturbed gene list.

---

---

**Drug Response Prediction Task**

**System Prompt:**

You are a helpful biology expert. You are an expert who knows a lot about single cell biology and genomics and will help me solve a series of tasks related to single cell data analysis.

**Guidance**

- If there is provided gene lists, they represent the top {n_genes} highly expressed genes of each cell, ranked in descending order. These serve as representative expression profiles for the cells and can be used to infer their likely identities.
- Based on your expertise, decide independently which perspectives (e.g., marker genes, tissue architecture, cell-cell interactions) are most informative for this analysis.

**Output Rules**

- Put your reasoning process inside `<analyze></analyze>` tags.
- Put your final decision inside `<answer></answer>` tags.

**User Prompt:**

Below is a list of {gene_names} ordered by descending expression level in a cell. Your task is to predict its response to the drug Erlotinib: [resistant, sensitive]. Make your choice in format `Final Answer:  resistant` or `Final Answer:  sensitive`.

{gene_names}: MALSU1 MT-TD MT-CYB MT-CO2 RBM4 MT-ND4L ...

---

**Condition Cell Generation**

**System Prompt:**

**Task**

- **Goal**: Generate a pseudo cell by predicting the top-expressed genes in a specific cell type and organ context.
- **Context**: You will be provided with the following information:
    1. The organ from which the cell originates: {organ}.
    2. The cell type of interest: {cell_type}.
    3. The number of top highly expressed genes to generate: 200.
    4. The ground truth expression profile was derived from real single-cell RNA sequencing (scRNA-seq) data, where gene expression levels were averaged across cells of the same type and transformed using log1p. Your task is to simulate this profile by listing genes in descending order of expected expression.

**Guidance**

Use your biological knowledge to reason about the typical transcriptional signature of the specified cell type within the given organ.

**Output Rules**

- You should include your analysis in `<analyze></analyze>`.
- Your final answer should be enclosed in XML tags:

```
<answer>
1. gene_1
2. gene_2
3. gene_3
...
i. gene_i
...
200. gene_200
</answer>
```

- `gene_i` denotes the gene with the i-th highest expression in that cell; you should report this gene using the standard HGNC symbol.

**User Prompt:**

Your task is to generate a pseudo cell by outputting the gene expression rank in descending order.

- The cell is from organ: {organ}.
- The cell type is: {cell_type}.
- Generate the top 200 genes with the highest expression levels, sorted in descending order.
- Output using standard gene symbols.
- Ground truth was extracted from real scRNA-seq data, averaged across cells after log1p transformation.

**Gene Annotation**

**System Prompt:**

You are a specialized expert in Gene Ontology (GO) annotation and human molecular biology, with proficiency in retrieving and structuring accurate biological data.

**Task**

- **Goal**: Perform a comprehensive Gene Ontology (GO) annotation for a given human gene.
- **Context**: You will be provided with a single human gene Symbol (e.g., TP53).

**Guidance**

- Your annotation must be biologically accurate and consistent with up-to-date GO databases.
- The response must include the following information for the target gene:
    1. **Official Gene Name**: The full, official name.
    2. **Gene Type**: The biotype of the gene (e.g., `protein_coding`, `pseudogene`).
    3. **Summary**: A concise description of the gene's primary function and biological role.
    4. **GO (Gene Ontology)**: A detailed listing of the three subcategories:
        – **Molecular Function**
        – **Biological Process**
        – **Cellular Component**
- For each GO term listed, you must provide both its name and its official definition.
- If no reliable annotations are available for a specific GO category, explicitly state: `no reliable annotation is available`.

**Output Rules**

Place your annotation information inside `<answer></answer>` tags. No markdown formatting.

**User Prompt:**

Provide a full GO annotation for the human gene {gene}.

---

**Spatial Cell Type Annotation**

**System Prompt:**
You are a helpful biology expert. You are tasked with performing a **spatially-informed cell annotation task**. Based on the expression profiles (represented by lists of highly expressed genes) of a **center cell** and its **{N} neighboring cells**, you must infer the most probable type of the center cell.

**Task:**

- **Goal:** Given the expression profile of a center cell and its **{N}** spatial neighbors, identify the most probable cell type of the center.
- **Context:** You will be provided with the following information:
    1. One **center cell** with its top {n_genes} expressed genes.
    2. {N} **neighboring cells**, each represented by their own top expressed genes.
    3. All gene lists are obtained from **single-cell RNA sequencing**.
    4. The expression profiles will always follow the order: center cell followed by nearest neighbors.
    5. The neighboring cells are the closest cells to the center cell in spatial proximity.

You must select the answer from the following list:
{cell_type_list}

**Guidance:**

- Use your biological knowledge to reason about the potential relationships between the **center cell** and its **spatial neighbors**.
- Since neighboring cells are the closest in spatial distance, they may share microenvironmental or tissue-level context that should inform your prediction.
- The provided gene lists represent the top {n_genes} highly expressed genes of each cell, ranked in descending order. These serve as representative expression profiles for the cells and can be used to infer their likely identities.
- You should make full use of the spatial information while reasoning.
- Based on your expertise, decide independently which perspectives (e.g., marker genes, tissue architecture, cell-cell interactions) are most informative for this analysis.

**Output Rules:**

- Put your reasoning process inside `<analyze></analyze>` tags.
- Put your final decision inside `<answer></answer>` tags.
- The `<answer>` tag must contain one of the following formats: `<answer>{{cell_type}}</answer>`, where `{{cell_type}}` must exactly match one of the options in the provided list.

**User Prompt:**
Center cell (genes high → low):
{gene_0}

Neighbors:

- First neighbor: {gene_1}
- Second neighbor: {gene_2}

    ...

- {N}-th neighbor: {gene_{N}}

What is the type of the center cell?

