# OpenReview forum: "CeLLM: Can Large Language Models Achieve the AI  Virtual Cell ?"
_ICLR.cc/2026/Conference — ICLR 2026 Conference Desk Rejected Submission_

### Official Review · Reviewer_yQT6 · 2025-10-29

**Soundness:** 2
**Presentation:** 2
**Contribution:** 1
**Rating:** 2
**Confidence:** 4

**Summary:**

The paper introduces CeLLM, a benchmark for assessing large language models on cellular biology tasks with the long-term goal of informing an AI “virtual cell.” The work is motivated by limits in current single-cell foundation models, including narrow modality coverage, weak causal reasoning, and limited interpretability. The authors argue that LLMs can bridge heterogeneous modalities and provide natural-language reasoning, but that rigorous and broad evaluation is missing. CeLLM spans gene, cell, and omics levels, defines multiple task settings and metrics, and evaluates 15 models across open-source, proprietary, and biology-specific systems. The framework emphasizes robustness to input phrasing and prompting to enable apples-to-apples comparisons and reproducible tracking over time. Overall, CeLLM positions itself as a cross-scale resource that surfaces current gaps and aims to accelerate LLM methods for virtual-cell modeling.

**Strengths:**

- Broad coverage of related work and relevant biological tasks.
- A diverse panel of LLMs is evaluated, including open, closed, and biology-tuned systems, which provides a useful snapshot of the field.

**Weaknesses:**

- The main results compare LLMs to one another, but not to established task-specific methods. Without such baselines, it is difficult to judge whether LLMs meaningfully advance the state of the art for “virtual cell” capabilities. Please include representative baselines per task, for example CellTypist for cell annotation and foundation models such as scGPT for annotation and gene-perturbation prediction. Reporting both absolute performance and deltas would clarify where LLMs help.
- Much of the manuscript centers on prompt design and model ranking. It is not yet clear how the benchmark reveals biological mechanisms, suggests new hypotheses, or guides experimental follow-up. Consider adding case studies that connect model outputs to potential biological discoveries (e.g., proposing gene candidates, prioritizing perturbations, or explaining pathway-level effects) and analyzing when LLM reasoning aligns with known biology.
- Several settings are underspecified. For each task, please clearly state the dataset, input and output formats, train or evaluation splits, pre- and post-processing, metric definitions, and any filtering. Document prompt templates, sampling parameters, and the number of seeds. This will improve reproducibility and allow fair comparisons.
- Some conclusions in the main text appear to rely on tables in the appendix, but the links are not explicit. Add precise cross-references to the relevant tables or figures and, where helpful, bring key results into the main paper for readability.

**Questions:**

* How can the proposed benchmark guide biological research?
* Why Cell2Sentence always produces results with all zero metrics? Can you provide some example output of it?

---

> ### Author Response · Authors · 2025-11-23
> **Response to Reviewer yQT6 (1/3)**
>
> *We thank the reviewer for the valuable feedback. We have updated the main text and appendix according to your suggestions (Marked by blue).  Below, we address the reviewer’s concerns one by one, denoting weaknesses as ‘W’ and questions as ‘Q’.*
>
> > **W1:** The main results compare LLMs to one another, but not to established task-specific methods. Without such baselines, it is difficult to judge whether LLMs meaningfully advance the state of the art for “virtual cell” capabilities. Please include representative baselines per task, for example CellTypist for cell annotation and foundation models such as scGPT for annotation and gene-perturbation prediction. Reporting both absolute performance and deltas would clarify where LLMs help.
>
> **A1:** We thank the reviewer for emphasizing the importance of comparing LLMs to established task-specific baselines. Compared to specialized single-cell models, **a key advantage of LLMs is that they can, without task-specific fine-tuning, leverage broad prior knowledge to handle downstream tasks in a zero-shot manner** while also providing interpretable reasoning traces.
>
> To address the reviewer’s concern, we have added a zero-shot comparison between a representative single-cell foundation model, scGPT, and LLM (GPT-5) on the cell-type annotation task. As shown in Table Re1, **GPT-5 substantially outperforms scGPT in the zero-shot setting on all three datasets**:
>
> Table Re1. Zero-shot cell annotation accuracy(%) comparison between scGPT and GPT-5.
>
> | Model / Dataset   | Brain  | Myeloid | Organoid |
> | ----------------- | ------ | ------- | -------- |
> | scGPT             | 24.81  | 10.72   | 21.69    |
> | GPT-5             | 67.50  | 20.00   | 71.50    |
> | Performance Delta | +42.69 | +9.28   | +49.81   |
>
> These results are consistent with prior findings that single-cell foundation models perform poorly in zero-shot cell-type annotation [1], and show that  LLMs can  provide strong “virtual cell” capabilities even without access to task-specific labels. **In the supervised learning regime**, our experiments focus on LLM-based models such as Cell2Sentence[2] and  rBio[3], which, as reported in their original papers, **outperform strong domain-specific baselines** including scGPT[4], scGen[5], Gears[6], CellOT[7], etc. on annotation and perturbation-related tasks.
>
> Taken together, both in data-limited zero-shot scenarios and in supervised settings, these comparisons indicate that LLM-based approaches meaningfully advance the state of the art for “virtual cell” capabilities.
>
>
>
> > **W2:**  Much of the manuscript centers on prompt design and model ranking. It is not yet clear how the benchmark reveals biological mechanisms, suggests new hypotheses, or guides experimental follow-up. Consider adding case studies that connect model outputs to potential biological discoveries (e.g., proposing gene candidates, prioritizing perturbations, or explaining pathway-level effects) and analyzing when LLM reasoning aligns with known biology.
>
> **A2:** We thank the reviewer for the suggestion encouraging us to better connect CeLLM with concrete biological insights. **We have added a detailed case study in $\underline{\text{Appendix R.9 (ref: page 20-21, line 1069-1125)}}$**, illustrating both successful and failed outputs of the LLM (Qwen3-14B) on the brain dataset for the cell-type annotation task. As shown in the example in $\underline{\text{Appendix R.9.1(ref: page 20-21, line 1069-1102)}}$, the model overestimated the specificity of several transcription/splicing-related proteins and underestimated classical immune/epithelial markers, leading to incorrect reasoning. In contrast, in the example in $\underline{\text{Appendix R.9.2(ref: page 21,line 1104-1124)}}$, the LLM successfully captured truly informative “module-level signals” (e.g., cilia-related gene modules, myelination gene modules), resulting in correct predictions.
>
> These examples suggest that the LLM’s reasoning aligns more closely with established biological knowledge in two scenarios:
>
> - **When there exists a set of “textbook-level,” highly specific markers that appear together.**
>    In such cases, the model has encountered many sentences in its training corpus stating that “X is a marker for cell type Y,” forming a very clear pattern. Its inferences are therefore generally reliable.
>
> - **When the signal emerges at the “gene-set level” rather than from single genes.**
>    When multiple functionally related genes co-occur (e.g., a large set of cilia-associated genes), the LLM can detect this higher-order “gene-module” signal rather than relying solely on individual genes, reflecting deeper functional understanding.

---

> ### Author Response · Authors · 2025-11-23
> **Response to Reviewer yQT6 (2/3)**
>
> > **W3:** Several settings are underspecified. For each task, please clearly state the dataset, input and output formats, train or evaluation splits, pre- and post-processing, metric definitions, and any filtering. Document prompt templates, sampling parameters, and the number of seeds. This will improve reproducibility and allow fair comparisons.
>
> **A3:** We thank the reviewer for this extremely valuable suggestion. We fully agree that detailed descriptions of the experimental setup are essential for ensuring the reproducibility of our benchmark and enabling fair comparisons. **We have  thoroughly organized and documented the datasets, metrics, and other relevant information for each task in $\underline{\text{Appendix R.6 (page 18-19, line 918-1007)}}$**.
>
>
>
> > **W4:** Some conclusions in the main text appear to rely on tables in the appendix, but the links are not explicit. Add precise cross-references to the relevant tables or figures and, where helpful, bring key results into the main paper for readability.
>
> **A4:** We appreciate the reviewer's meticulous review and suggestion regarding clarity and traceability. We have thoroughly revised the main text to ensure that all conclusions derived from the experimental results in the main text include precise cross-references to the relevant tables in Appendix.
>
>
>
> > **Q1:** How can the proposed benchmark guide biological research?
>
> **A5:** We thank the reviewer for raising this excellent question regarding the practical utility of CeLLM in guiding and accelerating biological research. Broadly, CeLLM provides guidance to the biological research in the following three key areas:
>
> - **Model selection and risk assessment:** By revealing which LLMs are relatively reliable for which types of questions, CeLLM helps researchers decide when a model can be safely used as a research tool.
>
> - **Turning benchmark tasks directly into in silico screening tools:** Many tasks in CeLLM correspond directly to common experimental questions, such as “What happens if gene X is perturbed or drug Y is applied?” When a model performs well on these benchmark instances, researchers can use the same task interface on their own new data—effectively transforming CeLLM into a standardized single-cell “virtual screening panel.” Notably, the Cell2Sentence model evaluated in the CeLLM benchmark has already completed a full *in silico*-to-*in vitro* loop in a real tumor-immune context: in a virtual screen of over 4,000 compounds, it proposed the novel combination of a “CK2 inhibitor (silmitasertib) plus low-dose interferon,” which was subsequently validated in human neuroendocrine cells that were not part of the training set. The combination synergistically enhanced antigen-presentation levels by ~50%, demonstrating the model’s ability to generate concrete, testable hypotheses that can directly proceed to wet-lab validation.
>
> - **Providing interpretable reasoning:** Because the benchmark is designed around LLMs, models output not only predictions but also text-based reasoning. By comparing these explanations with established biological knowledge, researchers can quickly distinguish which suggestions are mechanistically plausible and which are more likely “hallucinations,” thereby obtaining both answers and confidence estimates.

---

> ### Author Response · Authors · 2025-11-23
> **Response to Reviewer yQT6 (3/3)**
>
> > **Q2:** Why Cell2Sentence always produces results with all zero metrics? Can you provide some example output of it?
>
> **A6:** We thank the reviewer for careful examination. The notably poor performance of Cell2Sentence (C2S) in our benchmark is an important observation, and our investigation reveals two main contributing factors, which also provide a crucial insight into the robustness of different training paradigms.
>
> **Sensitivity due to the SFT Training Paradigm.** For the sake of fair comparison, all models were evaluated using a unified set of prompt templates across the benchmark. However, **C2S was trained using a Supervised Fine-Tuning (SFT) paradigm, which heavily relies on the specific prompt used during its training phase.** This training approach appears to have caused catastrophic forgetting of the model's foundational knowledge, leading to difficulties in providing correct answers for new tasks or when presented with different prompting styles. **Examples of C2S's output  can be found in $\underline{\text{Appendix R.8 (ref: page 19-20, line 1022-1068)}}$.** In contrast, Cell-o1, which was trained on the cell puzzle task using a Reinforcement Learning (RL) paradigm, did not exhibit this issue of foundational knowledge forgetting. **This comparison highlights a significant difference in robustness stemming from the choice of training paradigm (SFT vs. RL)**.
>
> **Poor Prompt Robustness.** To further investigate its limitations, **we test C2S using the model's officially provided cell type annotation prompt on the cell annotation task.** The results, presented in Table Re2, show that C2S's performance remained poor compared to other models, further confirming its low robustness to varied prompting in a general setting.
>
> Table Re2. Single-cell annotation accuracy (%) comparison under different prompt.
>
> | Model / Dataset        | Brain | Myeloid | Organoid |
> | ---------------------- | ----- | ------- | -------- |
> | C2S (CeLLM prompt)     | 0     | 0       | 0        |
> | C2S  (C2S prompt)      | 58.50 | 25.00   | 32.50    |
> | Cell-o1 (CeLLM prompt) | 47.00 | 21.00   | 23.50    |
> | Cell-o1 (C2S prompt)   | 46.50 | 22.00   | 21.00    |
> | GPT-5   (CeLLM prompt) | 67.50 | 20.00   | 71.50    |
> | GPT-5   (C2S prompt)   | 67.00 | 20.00   | 71.00    |
>
> This experimental contrast is a valuable finding: it demonstrates that the SFT and RL training paradigms  lead to differing levels of robustness in downstream tasks when the model is faced with new problems or prompts.
>
>
>
> **Reference**
>
> [1] Zero‑shot evaluation reveals limitations of single‑cell foundation models.
>
> [2] Scaling Large Language Models for Next-Generation Single-Cell Analysis.
>
> [3] rbio1 - training scientific reasoning LLMs with biological world models as soft verifiers.
>
> [4] scGPT: toward building a foundation model for single-cell multi-omics using generative AI.
>
> [5] scGen predicts single-cell perturbation responses.
>
> [6] Predicting transcriptional outcomes of novel multigene perturbations with GEARS.
>
> [7] Learning single-cell perturbation responses using neural optimal transport.
>
> ---
>
> *We greatly appreciate your time and effort in reviewing our manuscript and providing helpful comments, as they will undoubtedly help us improve the quality of our article. If our response has successfully addressed your concerns and clarified any ambiguities, we respectfully hope that you consider raising the score. Should you have any further questions or require additional clarification, we would be delighted to engage in further discussion.*

---

### Official Review · Reviewer_jCwd · 2025-10-31

**Soundness:** 3
**Presentation:** 3
**Contribution:** 2
**Rating:** 4
**Confidence:** 4

**Summary:**

CeLLM presents a benchmark for evaluating large language models (LLMs) in cellular biology. It aggregates existing datasets across gene, cell, and omics-level tasks, standardizes evaluation protocols, and tests a range of general and domain-specific LLMs. The benchmark includes multiple task types, prompt-format studies, and robustness analyses (e.g., gene-order perturbations). The paper uses three categories of evaluation metrics classification, ranking, and foundation model (FM)-as-judge.

Recommendation: (leaning reject)
CeLLM is a well-executed, comprehensive evaluation study that will be useful for practitioners and newcomers to LLMs in biology. However, its scientific novelty and methodological depth are limited. The paper’s value is primarily as a resource and infrastructure contribution, which might better suit a datasets/benchmarks venue or a companion release rather than a main-track research paper at ICLR.

Supporting arguments
1. The study adds organizational and empirical clarity but not new methods or theoretical insights.
2. The use of FM-as-judge (especially with Geneformer) is interesting but insufficiently documented and should include main-text results.
3. Representing cells as sorted gene tokens oversimplifies biology and limits generalization, though it aligns with prior LLM-pretraining conventions. Many biomedical foundation models (e.g. scGPT) use more better data structure to capture more biomedical meaning, and should be added to the comparison. As is, all models tested are text-only LLMs, which is a very limited/biased type of model for Virtual Cell.
4. Despite these limitations, the work’s reproducibility and clarity make it a potentially valuable reference for future research benchmarking.

**Strengths:**

1. Comprehensive and systematic evaluation. The study is well organized and clearly written, offering a large-scale and reproducible comparison of existing LLMs on biological tasks.

2. Practical contribution. While not conceptually novel, the work is valuable for practitioners who lack the resources to replicate such broad comparisons. It highlights prompt design effects, scaling trends, and robustness issues in a consistent setting.

3. Reproducibility and structure. The standardized preprocessing, task definitions, and metrics make this a useful reference for future model developers.

4. Breadth of coverage. The inclusion of multiple biological task types (gene-level, cell-level, cross-omics) provides a wide view of LLM performance across domains.

**Weaknesses:**

1. Limited novelty. The paper mainly evaluates existing models on existing datasets with standard metrics. Its contribution is systematic rather than conceptual or methodological. This limits its fit for the main ICLR track, though it would be valuable as a benchmark paper.

2. Simplistic cell representation. Cells are represented as sorted lists of genes, which is a narrow and biologically limited representation that neglects expression magnitudes, gene–gene dependencies, and spatial or pathway-level structure. (note that this is a limitation of any LLM that only takes text modality as input; this can be a deal breaker for even using text-only LLM as a candidate for Virtual Cell, which required a lot more meaningful representation)

3. FM-as-judge clarity and placement. While FM-as-judge evaluation is a standard idea, the paper’s implementation using Geneformer as a judge is not sufficiently detailed, and results are buried or absent from the main text. It’s unclear how scores were computed or how they complement traditional metrics. Clearer explanations and summary results in the main body would strengthen the paper.

4. Overall impression. The paper reads largely as a series of ablation studies on known models, datasets, and prompt styles. Although useful for the field, it offers limited new insights or conceptual advances.

**Questions:**

1. Could you elaborate on how Geneformer is used as a judge? For example, what layer or embedding is used, how are scores computed, and how does this correlate with traditional metrics?

2. Have you considered richer or more biologically grounded cell representations in addition to Geneformer style sorted list of genes? Maybe compare to those models using the same tasks and explore and explain the limitation of the sorted gene based cell representation?

3. How do you ensure that evaluated models have not seen parts of the benchmark data during pretraining? Some LLMs, especially the larger ones, may have the unfair advantage of seeing the benchmark data during training.

**Details Of Ethics Concerns:**

No concern.

---

> ### Author Response · Authors · 2025-11-23
> **Response to Reviewer jCwd (1/4)**
>
> *We thank the reviewer for the valuable feedback. We have updated the main text and appendix according to your suggestions (Marked by blue).  Below, we address the reviewer’s concerns one by one, denoting weaknesses as ‘W’ and questions as ‘Q’.*
>
> > **W1:** Limited novelty. The paper mainly evaluates existing models on existing datasets with standard metrics. Its contribution is systematic rather than conceptual or methodological. This limits its fit for the main ICLR track, though it would be valuable as a benchmark paper.
>
> **A1:** We appreciate the reviewer's assessment of our work. **We would like to kindly remind the reviewer that this paper was submitted to the Datasets and Benchmarks area at ICLR 2026.** As a benchmark paper, the novelty of CeLLM is primarily reflected in two major aspects:
>
> 1. **Comprehensive Scope and Experimental Setup:** We designed CeLLM for a holistic evaluation of the latest advances in Cell LLMs by incorporating:
>
>    - **New Datasets & Task Diversity:** A wide array of tasks spanning multiple biological levels (gene, omics, and cell) using biological text datasets constructed by us.
>
>    - **Model Coverage:** The most up-to-date and extensive collection of models, including open-source, closed-source, and domain-specific LLMs.
>    - **In-depth Analysis:** Comprehensive exploration of crucial factors influencing task performance.
>    - **Metrics:** A full spectrum of evaluation metrics to thoroughly assess model capabilities.
>
> 2. **Novel Insights and Actionable Takeaways:** Our extensive analysis, including both fundamental benchmark results and detailed ablation studies , yields significant, important insights and actionable takeaways **(summarized  in $\underline{\text{Appendix R.3 (ref: page15-16, line787-858)}}$)**.
>
>
>
> > **W2:** Simplistic cell representation. Cells are represented as sorted lists of genes, which is a narrow and biologically limited representation that neglects expression magnitudes, gene–gene dependencies, and spatial or pathway-level structure. (note that this is a limitation of any LLM that only takes text modality as input; this can be a deal breaker for even using text-only LLM as a candidate for Virtual Cell, which required a lot more meaningful representation)
>
> **A2:** We sincerely appreciate the reviewer’s insightful comments regarding current cell representations. Our choice to use the sorted lists of genes (cell sentence) format in the benchmark is motivated by three key reasons:
>
> **Alignment with the prevailing paradigm and benchmark focus:** The “cell sentence” representation is the most widely adopted format in state-of-the-art Cell LLMs. For example, both the Cell2Sentence and Cell-o1 models use this approach for downstream tasks. Since CeLLM aims to evaluate progress within this emerging paradigm, adopting this mainstream representation is essential.
>
> **Information retention and interchangeability:** Although the representation is textual, it preserves critical biological information. The Cell2Sentence paper [1] demonstrated a strong linear relationship between gene order in the cell sentence and gene expression levels in log space ($R^{2}$=0.85). This indicates that very little information is lost during the conversion. Such interchangeability allows leveraging the strengths of LLMs in natural language processing while retaining the ability to convert the representation back into gene-expression vectors for use in traditional single-cell analysis methods.
>
> **Robustness:** Furthermore, work related to cell text representations [2] highlights that the ranking mechanism serves as a form of rank normalization. This property enhances the model’s robustness to noise, enabling more stable performance when processing real-world single-cell data.

---

> ### Author Response · Authors · 2025-11-23
> **Response to Reviewer jCwd (2/4)**
>
> > **W3:** FM-as-judge clarity and placement. While FM-as-judge evaluation is a standard idea, the paper’s implementation using Geneformer as a judge is not sufficiently detailed, and results are buried or absent from the main text. It’s unclear how scores were computed or how they complement traditional metrics. Clearer explanations and summary results in the main body would strengthen the paper.
>
> > **Q1 :** Could you elaborate on how Geneformer is used as a judge? For example, what layer or embedding is used, how are scores computed, and how does this correlate with traditional metrics?
>
> **A3:** We completely agree with the reviewers on the necessity of providing a clearer and more detailed exposition of the Geneformer-as-a-Judge evaluation methodology.  **We added an explanation in the $\underline{\text{main text (ref: page 6, line 301-306)}}$ clarifying the significance of Geneformer cosine similarity compared with traditional metrics. The detailed computation procedure is provided in $\underline{\text{Appendix A2.3(ref: page 23, line 1215-1241)}}$, and the “Cosine sim.” columns in $\underline{\text{Tables 15(ref: page 29, line 1543-1562) and 17(ref: page 30, line 1597-1617)}}$ report the Geneformer cosine similarity values for different generation tasks.**
>
> In the Cell Generation and Perturbed Cell Generation tasks, we require the model to output the top-k genes ranked by expression level. We use Geneformer-V2-316M as the evaluator. Given a cell-sentence gene list as input, genes that are not included in the Geneformer tokenizer are filtered out during computation. We extract the CLS token vector from the final Transformer layer of Geneformer and treat it as the cell representation. **We compute the cosine similarity between the Geneformer representations of the real cell sentence and the LLM-generated cell sentence.**
>
> Traditional metrics measure "whether the gene orderings are similar“, whereas Geneformer cosine similarity measures whether “the generated cells lie in the correct biological state.” **Together, they form a complementary evaluation framework that spans both low-dimensional and high-dimensional feature spaces, integrating explicit structural information with implicit biological semantics.**
>
>
>
> > **W4:** Overall impression. The paper reads largely as a series of ablation studies on known models, datasets, and prompt styles. Although useful for the field, it offers limited new insights or conceptual advances
>
> **A4:** We appreciate the reviewers for highlighting this critical point. We fully agree that translating our extensive experimental results into clear, actionable conclusions are vital for advancing the field. **We have added a dedicated section, $\underline{\text{Appendix R.3 (ref: page15-16, line787-858)}}$, to summarize the key insights, actionable takeaways, limitations, and promising directions for future work.** This section synthesizes the higher-level observations across different tasks, providing concrete guidance for future research and development of Cell LLMs.
>
> Below is a brief summary of the most significant Insights and actionable takeaways derived from the CeLLM benchmark:
>
> **1.Model Choice vs. Size:** Model choice should be task-dependent rather than uniformly “bigger is better.”
>
> **2.Strategic Reasoning:** Reasoning (“think”) modes should be used selectively.
>
> **3.Input Construction:** Input construction for “cell sentences” should utilize medium gene lengths (100–200 genes) rather than naive long contexts.
>
> **4.Multi-Omics Integration:** Multi-omics data should not be concatenated naively; larger models exploit integration strategies more effectively.
>
> **5.Prompt Engineering:** Prompt framing and few-shot design critically shape model behavior.
>
> **6.Spatial Information Encoding:** Spatial information must be encoded carefully.
>
> **7.Evaluation Design:** Evaluation should combine classic metrics (e.g., F1, AUC) with Foundation Model-as-a-Judge (FM-as-judge) metrics for biological realism. This hybrid approach ensures both statistical fidelity and biological relevance.

---

> ### Author Response · Authors · 2025-11-23
> **Response to Reviewer jCwd (3/4)**
>
> > **Q2:** Have you considered richer or more biologically grounded cell representations in addition to Geneformer style sorted list of genes? Maybe compare to those models using the same tasks and explore and explain the limitation of the sorted gene based cell representation?
>
> **A5:** We appreciate the reviewer’s comments regarding the need for richer cellular text representations. We observed that when given the input *cell sentence*, LLMs can automatically infer pathway-related information from the gene list in downstream task ($\underline{\text{Appendix R.4 (ref: page 17, line 878-887)}}$). In addition, to examine other types of cell representations, we tested four representative models **using different variants of the cell sentence: the original version, version augmented with expression information, and version incorporating gene co-expression information.** The results on the cell-type annotation task are shown in Table Re1.
>
> Table Re1. Performance comparision(%) of different cell representations on cell-type annotation.
>
> | Model     | cell sentence | with expression | with  gene co-expression |
> | --------- | ------------- | --------------- | ------------------------ |
> | Qwen3-8B  | **72.14**     | 72.04           | 66.17                    |
> | Qwen3-32B | **84.08**     | 83.08           | 77.61                    |
> | Cell-O1   | **38.81**     | 25.37           | 34.83                    |
> | GPT-5     | **89.55**     | 89.55           | 89.55                    |
>
> As the table indicates, **providing additional information did not yield further performance gains; on the contrary, it even negatively affected the performance of smaller models.**
>
> Regarding spatial information, in the spatial cell-annotation task in our paper, we conveyed spatial context to the LLM by constructing structured multiple cell sentences based on the distances of neighboring cells to the center cell. We found that not all models were able to make effective use of this spatial information. **Therefore, for spatial transcriptomics task, the current cell sentence format still has limitation.** Designing more effective and generalizable spatial cell-encoding strategies remains an important direction for future exploration.

---

> ### Author Response · Authors · 2025-11-23
> **Response to Reviewer jCwd (4/4)**
>
> > **Q3:** How do you ensure that evaluated models have not seen parts of the benchmark data during pretraining? Some LLMs, especially the larger ones, may have the unfair advantage of seeing the benchmark data during training.
>
> **A6:** We thank the reviewer for raising the important concern that the evaluation models may have seen the benchmark data during pre-training. **We verify that the benchmark datasets poses no issues regarding fair evaluation of different LLMs on downstream tasks.**
>
> **Data provenance and novel textualization.** Except for the gene-annotation task, all CeLLM tasks are constructed directly from primary single-cell or spatial omics matrices (scRNA-seq, CITE-seq, spatial transcriptomics). These raw count matrices are not natural-language documents and, to the best of our knowledge, are not included in any LLM pretraining corpora. Moreover, the inputs used in CeLLM are produced through our newly designed cell-sentence and multi-omics-sentence textualization pipeline, which serializes high-dimensional omics vectors into descriptive sentences. These templates and serialization rules did not exist prior to CeLLM, making it impossible for any LLM to have encountered identical input–output pairs during pretraining.
>
> **Gene-annotation task and overlap with biological text.** For the gene-annotation task, we intentionally use Gene-Ontology–derived descriptions. These may partially overlap with public biological text, but this is by design: the goal is to evaluate whether an LLM can function as a gene-centric knowledge base. Exposure to biological corpora during pretraining is therefore part of the evaluated capability rather than an unfair advantage.
>
> **Domain-specific models.** For domain-adapted models such as rBio, Cell-o1, and C2S, which are trained on textualized omics data, we ensure that our CeLLM test datasets do not overlap with the datasets used for training in their respective papers, thereby eliminating dataset-level leakage.
>
> **Empirical evidence against contamination advantages.** Across tasks such as spatial cell annotation and differential-gene prediction, larger models do not consistently outperform smaller models. This pattern indicates that CeLLM tasks require genuine biological reasoning rather than memorization of any potential pretraining data, providing additional evidence that contamination is unlikely to influence our conclusions.
>
> Thank you for providing this critical question！ We have added the above discussion to $\underline{\text{Appendix R.5(ref: page 17-18,line 909-958)}}$ to improve our paper.
>
>
>
> **Reference**
>
> [1] Scaling Large Language Models for Next-Generation Single-Cell Analysis.
>
> [2] Representing cells as sentences enables natural-language processing for single-cell transcriptomics.
>
> ---
>
> *We greatly appreciate your time and effort in reviewing our manuscript and providing helpful comments, as they will undoubtedly help us improve the quality of our article. If our response has successfully addressed your concerns and clarified any ambiguities, we respectfully hope that you consider raising the score. Should you have any further questions or require additional clarification, we would be delighted to engage in further discussion.*

---

### Official Review · Reviewer_J6jt · 2025-10-31

**Soundness:** 3
**Presentation:** 3
**Contribution:** 2
**Rating:** 6
**Confidence:** 4

**Summary:**

This work introduces CeLLM, a unified benchmark for evaluating LLMs in the cellular domain. CeLLM notably includes a broad spectrum of tasks, and incorporates diverse evaluation criteria that go beyond accuracy, F1-score, and related measures. In this study, 15 models are evaluated on this benchmark, including open-source and proprietary general LLMs (ie. GPT-5, DeepSeek-V3.1), and domain-specific LLMs (ie. cell-o1). This work also provides an investigation of key factors influencing LLM performance on cellular tasks.

**Strengths:**

•	Addresses an important and clearly articulated gap in the literature, as the area of foundation models and LLMs for single-cell analysis has seen an influx of models and methods but insufficient benchmarks to properly evaluate them.
•	Diverse range of tasks and models evaluated, with the former in particular contributing to the novelty of this work and its potential for impact.
•	Selection of metrics is well thought out and represents a good spectrum of task difficulties and topic areas.
•	Investigation of key factors influencing LLM performance is interesting.
•	Main conclusions are non-trivial and useful.
•	Writing is generally clear.

**Weaknesses:**

•	Unclear whether models are evaluated fairly. For example, the authors reported an accuracy of 0% for Cell2Sentence across nearly all tasks, including cell type annotation, which the model was able to perform reasonably well in the original work (https://pmc.ncbi.nlm.nih.gov/articles/PMC11565894/pdf/nihpp-2023.09.11.557287v4.pdf), and therefore has at least some capability in this setting. Ultimately, this benchmark may just be evaluating how knowledgeable these models are in cellular biology when prompted and evaluated in a specific way, rather than being a proper comprehensive evaluation of their knowledge and capabilities.
•	Limited actionable takeaways. Comparisons of different models often don’t lead to concrete conclusions/recommendations. Higher level observations across different tasks not significantly discussed. Limitations of this work and potential future work are not clearly outlined.

Ultimately, despite the benchmark proposed being interesting, these weaknesses limit this work’s potential for impact. I am therefore providing an initial recommendation of marginal accept. However, if the authors are able to better justify some of their choices, and provide more concrete conclusions, I believe this work could meet the bar for ICLR.

**Questions:**

•	Why is the accuracy of Cell2Sentence so poor on the benchmark? Is it related to how the problem was framed, or is there a notable difference between the benchmark and the benchmarks used in the Cell2Sentence paper?
•	Can a more detailed discussion of actionable takeaways be provided?

Some suggestions:
•	Consider bolding the highest values for different metrics in your tables to improve readability.
•	Consider the use of third level headings in section 3 (i.e.. I believe “Gene Annotation” should be 7.2.1 since it is under Main Results).

---

> ### Author Response · Authors · 2025-11-23
> **Response to Reviewer J6jt （1/2）**
>
> *We thank the reviewer for the valuable feedback. We are glad that the reviewer found our contributions valuable.We have updated the main text and appendix according to your suggestions (Marked by blue).  Below, we address the reviewer’s concerns one by one, denoting weaknesses as ‘W’ and questions as ‘Q’.*
>
> > **W1:** Unclear whether models are evaluated fairly. For example, the authors reported an accuracy of 0% for Cell2Sentence across nearly all tasks, including cell type annotation, which the model was able to perform reasonably well in the original work (https://pmc.ncbi.nlm.nih.gov/articles/PMC11565894/pdf/nihpp-2023.09.11.557287v4.pdf), and therefore has at least some capability in this setting. Ultimately, this benchmark may just be evaluating how knowledgeable these models are in cellular biology when prompted and evaluated in a specific way, rather than being a proper comprehensive evaluation of their knowledge and capabilities.
>
> > **Q1:** Why is the accuracy of Cell2Sentence so poor on the benchmark? Is it related to how the problem was framed, or is there a notable difference between the benchmark and the benchmarks used in the Cell2Sentence paper?
>
> **A1:** We thank the reviewer for careful examination. The notably poor performance of Cell2Sentence (C2S) in our benchmark is an important observation, and our investigation reveals two main contributing factors, which also provide a crucial insight into the robustness of different training paradigms.
>
> **Sensitivity due to the SFT Training Paradigm.** For the sake of fair comparison, all models were evaluated using a unified set of prompt templates across the benchmark. However, **C2S was trained using a Supervised Fine-Tuning (SFT) paradigm, which heavily relies on the specific prompt used during its training phase.** This training approach appears to have caused catastrophic forgetting of the model's foundational knowledge, leading to difficulties in providing correct answers for new tasks or when presented with different prompting styles. **Examples of C2S's output  can be found in $\underline{\text{Appendix R.8 (ref: page 19-20, line 1022-1068)}}$.** In contrast, Cell-o1, which was trained on the cell puzzle task using a Reinforcement Learning (RL) paradigm, did not exhibit this issue of foundational knowledge forgetting. **This comparison highlights a significant difference in robustness stemming from the choice of training paradigm (SFT vs. RL)**.
>
> **Poor Prompt Robustness.** To further investigate its limitations, **we test C2S using the model's officially provided cell type annotation prompt on the cell annotation task.** The results, presented in Table Re1, show that C2S's performance remained poor compared to other models, further confirming its low robustness to varied prompting in a general setting.
>
> Table Re1. Single-cell annotation accuracy (%) comparison under different prompt.
>
> | Model / Dataset        | Brain | Myeloid | Organoid |
> | ---------------------- | ----- | ------- | -------- |
> | C2S (CeLLM prompt)     | 0     | 0       | 0        |
> | C2S  (C2S prompt)      | 58.50 | 25.00   | 32.50    |
> | Cell-o1 (CeLLM prompt) | 47.00 | 21.00   | 23.50    |
> | Cell-o1 (C2S prompt)   | 46.50 | 22.00   | 21.00    |
> | GPT-5   (CeLLM prompt) | 67.50 | 20.00   | 71.50    |
> | GPT-5   (C2S prompt)   | 67.00 | 20.00   | 71.00    |
>
> This experimental contrast is a valuable finding: it demonstrates that the SFT and RL training paradigms  lead to differing levels of robustness in downstream tasks when the model is faced with new problems or prompts.

---

> ### Author Response · Authors · 2025-11-23
> **Response to Reviewer J6jt （2/2）**
>
> > **W2:** Limited actionable takeaways. Comparisons of different models often don’t lead to concrete conclusions/recommendations. Higher level observations across different tasks not significantly discussed. Limitations of this work and potential future work are not clearly outlined.
>
> > **Q2:**  Can a more detailed discussion of actionable takeaways be provided?
>
> **A2:** We appreciate the reviewers for highlighting this critical point. We fully agree that translating our extensive experimental results into clear, actionable conclusions and outlining the work's limitations are vital for advancing the field. **We have added a dedicated section, $\underline{\text{Appendix R.3 (ref: page15-16, line787-858)}}$, to summarize the key insights, actionable takeaways, limitations, and promising directions for future work.** This section synthesizes the higher-level observations across different tasks, providing concrete guidance for future research and development of Cell LLMs.
>
> Below is a brief summary of the most significant Insights and actionable takeaways derived from the CeLLM benchmark:
>
> **1.Model Choice vs. Size:** Model choice should be task-dependent rather than uniformly “bigger is better.”
>
> **2.Strategic Reasoning:** Reasoning (“think”) modes should be used selectively.
>
> **3.Input Construction:** Input construction for “cell sentences” should utilize medium gene lengths (100–200 genes) rather than naive long contexts.
>
> **4.Multi-Omics Integration:** Multi-omics data should not be concatenated naively; larger models exploit integration strategies more effectively.
>
> **5.Prompt Engineering:** Prompt framing and few-shot design critically shape model behavior.
>
> **6.Spatial Information Encoding:** Spatial information must be encoded carefully.
>
> **7.Evaluation Design:** Evaluation should combine classic metrics (e.g., F1, AUC) with Foundation Model-as-a-Judge (FM-as-judge) metrics for biological realism. This hybrid approach ensures both statistical fidelity and biological relevance.
>
>
>
> > **Q3:** Consider bolding the highest values for different metrics in your tables to improve readability.
>
> **A3:** We appreciate this helpful suggestion regarding table readability. Following the reviewer's advice, we have updated all tables throughout the manuscript and supplementary material to bold the highest performing value for each metric.
>
>
>
> > **Q4:** Consider the use of third level headings in section 3 (i.e.. I believe “Gene Annotation” should be 7.2.1 since it is under Main Results).
>
> **A4:** We appreciate the reviewer's meticulous review and helpful structural suggestion.  We agree that the hierarchical structure needed refinement. We have revised the section headings and now place all downstream task subsections (e.g., Gene Annotation, Cell Type Annotation) under a unified Section 7.2, adjusting the numbering accordingly.
>
> ---
>
> *We greatly appreciate your time and effort in reviewing our manuscript and providing helpful comments, as they will undoubtedly help us improve the quality of our article. If our response has successfully addressed your concerns and clarified any ambiguities, we respectfully hope that you consider raising the score. Should you have any further questions or require additional clarification, we would be delighted to engage in further discussion.*

---

### Official Review · Reviewer_5GKb · 2025-11-09

**Soundness:** 3
**Presentation:** 3
**Contribution:** 3
**Rating:** 6
**Confidence:** 4

**Summary:**

This paper introduces CeLLM, a unified benchmark for evaluating large language models (LLMs) on core cellular-biology tasks spanning genes, cells, and multi-omics. It assesses 15 open-source, closed-source, and domain-specialized models across diverse tasks (gene annotation, multi-omics integration, cell annotation—single, batch, spatial—cell perturbation, drug response, and cell-condition generation), with multiple evaluation criteria (classification, ranking, and foundation-model–based metrics). The benchmark aims to be cross-scale, reproducible, and evolving to support progress toward “Artificial Intelligence Virtual Cells” (AIVC).

**Strengths:**

Overall, CeLLM could be a significant benchmarking contribution for AI-for-cell biology, contingent on clarifications about novelty claims, data/preprocessing rigor, and ablation reporting. With those clarifications in an author response, I would be willing to increase the score.

**Weaknesses:**

1. The paper promises public code/data upon publication; consider releasing evaluation scripts and prompts (including few-shot exemplars) earlier to encourage community adoption.

2. The foundation-model–based evaluation (embedding similarity) is reasonable, but please discuss dependence on the chosen evaluator (e.g., Geneformer) and risks of circularity if any evaluated model is aligned to the same latent space.

**Questions:**

1. Please make the benchmark’s novelty against prior efforts more explicit in the main text: e.g., what’s new beyond existing task suites and datasets for cell annotation, perturbation, and multi-omics (CellVerse/OP tasks, etc.)? A short table contrasting CeLLM’s tasks/metrics/models with prior benchmarks would help readers understand incremental vs. new coverage.

2. The “optimal window” of 100–200 genes is interesting; please show full curves (20/50/100/200/500) with CIs across models and tasks. Likewise, for few-shot settings, report performance as a function of k-shots for both “leak” and “mask” strategies (the text notes leak helps, mask can hurt).

Typos:
1. “Biobert and ROGUE” → “BioBERT and ROUGE” (gene-annotation setup).
2. “From the table 14” → “From Table 14”.

---

> ### Author Response · Authors · 2025-11-23
> **Response to Reviewer 5GKb (1/3)**
>
> *We thank the reviewer for the valuable feedback. We are glad that the reviewer found our contributions valuable.We have updated the main text and appendix according to your suggestions (Marked by blue).  Below, we address the reviewer’s concerns one by one, denoting weaknesses as ‘W’ and questions as ‘Q’.*
>
> > **W1:** The paper promises public code/data upon publication; consider releasing evaluation scripts and prompts (including few-shot exemplars) earlier to encourage community adoption.
>
> **A1:** We sincerely thank the reviewer for the positive feedback and the constructive suggestion regarding the earlier release of our resources. We are pleased to confirm that we have already made the core implementation publicly available on the anonymous GitHub repository:https://anonymous.4open.science/r/CeLLM-ICLR2026. This release includes the main codebase, the data processing pipeline, the evaluation scripts, and the prompt templates (including the few-shot exemplars mentioned by the reviewer), which should facilitate early community access and adoption of our benchmark.
>
>
>
> > **W2:** The foundation-model–based evaluation (embedding similarity) is reasonable, but please discuss dependence on the chosen evaluator (e.g., Geneformer) and risks of circularity if any evaluated model is aligned to the same latent space.
>
> **A2:** This is a profound insight and a very important point regarding the robustness and objectivity of our evaluation methodology. We appreciate the opportunity to clarify this.
>
> **Dependence on Geneformer.** We chose Geneformer as the foundation-model-based evaluator for the following key reason:
>
> Geneformer is trained using rank-value encoding of gene usage (i.e., the order of gene expression) as input. **This input format aligns directly with the "cell sentence" modeling paradigm used in our cell generation tasks.** In contrast, most other established single-cell foundation models (e.g., scFoundation, scGPT) require gene expression values as input. This architectural similarity makes Geneformer the most convenient and suitable choice for robustly evaluating the generated textual gene sequences.
>
> **Risk of Circularity and Latent Space Alignment.** Crucially, the evaluated models (both general-purpose LLMs and specialized Cell-o1) and Geneformer do not share the same latent space, thereby mitigating the risk of circularity. This distinction stems from their fundamentally different training paradigms and objectives:
>
> - General-Purpose LLMs: The majority of the LLMs we tested (e.g., Gemini 2.5 Pro) are pretrained extensively on general text corpora and have not been exposed to training data in the Geneformer's specific gene rank-order format. Furthermore, LLM output on cell generation task in $\underline{\text{Appendix R.7(ref: page 19, line 1009-1021)}}$ suggests that  **LLM analyze characteristic genes and reason biologically before generating the sequence**. This indicates a utilization of their knowledge base for analytical reasoning, rather than merely fitting a sequence to a pre-existing latent space shared with Geneformer.
>
> - Cell-o1 Model: While Cell-o1 was trained using the "cell sentence" format, it employs an **autoregressive paradigm**, focused on generation. Conversely, **Geneformer utilizes a Masked Language Modeling (BERT-style) pretraining approach**, focused predominantly on comprehension and representation. These distinct objectives and architectures ensure non-alignment between their resulting latent spaces. Additionally, as shown in $\underline{\text{Table 17 (ref: page 30,line 1597-1617)}}$, Cell-o1's performance on the cell generation task is inferior to the strong performance of general LLMs, further supporting the conclusion that its specialized, autoregressive space does not fully align with the general comprehension space captured by Geneformer.

---

> ### Author Response · Authors · 2025-11-23
> **Response to Reviewer 5GKb (2/3)**
>
> > **Q1:** Please make the benchmark’s novelty against prior efforts more explicit in the main text: e.g., what’s new beyond existing task suites and datasets for cell annotation, perturbation, and multi-omics (CellVerse/OP tasks, etc.)? A short table contrasting CeLLM’s tasks/metrics/models with prior benchmarks would help readers understand incremental vs. new coverage.
>
> **A3:** We appreciate this highly constructive suggestion. Following the reviewer's suggestion, **we have explicitly detailed the novelty in a comparative table provided in $\underline{\text{Appendix R.1(ref: page 14, line 728-755)}}$**. This table contrasts CeLLM's tasks, metrics, and models with other recent natural language-centric benchmarks, specifically the Single-Cell Omics Arena [1] and CellVerse [2], to clearly illustrate our expanded coverage and advancements. The increasing prevalence of natural language-centric models for cellular problems (e.g., Cell-O1, C2S, rBio) necessitates a new, comprehensive evaluation suite. The novelty of CeLLM, compared to prior  single-cell benchmarks, is primarily reflected in two major aspects:
>
> 1. **Comprehensive Scope and Experimental Setup:** We designed CeLLM for a holistic evaluation of the latest advances in Cell LLMs by incorporating:
>
>    - **Task Diversity:** A wide array of tasks spanning multiple biological levels (gene, omics, and cell).
>
>    - **Model Coverage:** The most up-to-date and extensive collection of models, including open-source, closed-source, and domain-specific LLMs.
>    - **In-depth Analysis:** Comprehensive exploration of crucial factors influencing task performance.
>    - **Metrics:** A full spectrum of evaluation metrics to thoroughly assess model capabilities.
>
> 2. **Novel Insights and Actionable Takeaways:** Our extensive analysis, including both fundamental benchmark results and detailed ablation studies , yields significant, important insights and actionable takeaways (summarized  in $\underline{\text{Appendix R.3(ref: page15-16, line 787-858)}}$) that go beyond the conclusions drawn from previous single-cell benchmarks.
>
>
>
> > **Q2:** The “optimal window” of 100–200 genes is interesting; please show full curves (20/50/100/200/500) with CIs across models and tasks. Likewise, for few-shot settings, report performance as a function of k-shots for both “leak” and “mask” strategies (the text notes leak helps, mask can hurt).
>
> **A4:** We thank the reviewer for their careful examination and highly valuable suggestions. We have incorporated the requested comprehensive analyses into the $\underline{\text{Appendix R.2 (ref: page 15, line 756-772)}}$ to validate our findings and provide deeper insights.
>
>  **Analysis of Cell Sentence Length (Optimal Window).** As suggested, we have added detailed results in Appendix R.2.1 (ref: page 15, line 758-772). Figure 2 presents curves showing the performance of various models on the cell type annotation and perturbation DEG Classification tasks as a function of the cell sentence length. **The results confirm our initial findings**: performance across different LLMs generally increases with length, achieves an optimal effect at a length of 100 or 200 genes, and subsequently declines, strongly supporting the existence of an 'optimal window.'
>
> **Analysis of Few-Shot Strategies (k-shots).** The requested k-shot results are presented in $\underline{\text{Appendix R.2.2 (ref: page 15, line 774-785)}}$. Figure 3 displays the performance curves as a function of k-shots for both the 'Leak' and 'Mask' prompting strategies.
>
> - **Strategy Comparison:** We observe that the **'Leak' strategy consistently and significantly outperforms the 'Mask' strategy** across all tested values of k.
> - **Performance Trend:** Interestingly, **increasing the value of k** in the 'Leak' few-shot prompting strategy **does not necessarily lead to improved performance**; it causes a slight drop. We hypothesize that this is closely related to the **context handling capacity of the models**. For instance, large models like Qwen3-32B and GPT-5 show a smaller decline in performance than smaller models like Qwen-8B and Cell-o1. Designing better few-shot prompting techniques that can consistently leverage increased context to enhance LLM capabilities on cellular tasks remains a crucial and promising direction for future research.

---

> ### Author Response · Authors · 2025-11-23
> **Response to Reviewer 5GKb (3/3)**
>
> > **Q3:** “Biobert and ROGUE” → “BioBERT and ROUGE” (gene-annotation setup).
> >
> > “From the table 14” → “From Table 14”.
>
> **A5:** We sincerely thank the reviewer for careful reviewing and for catching these typos. We have corrected both instances in the main text:
>
> - "Biobert and ROGUE" has been updated to "BioBERT and ROUGE".
>
> - "From the table 14" has been corrected to "From Table 14".
>
>
>
> **Reference**
>
> [1] Single-Cell Omics Arena: A Benchmark Study for Large Language Models on Cell Type Annotation Using Single-Cell Data
>
> [2] CellVerse: Do Large Language Models Really Understand Cell Biology?
>
> ---
>
> *We greatly appreciate your time and effort in reviewing our manuscript and providing helpful comments, as they will undoubtedly help us improve the quality of our article. If our response has successfully addressed your concerns and clarified any ambiguities, we respectfully hope that you consider raising the score. Should you have any further questions or require additional clarification, we would be delighted to engage in further discussion.*

---

### Author Response · Authors · 2025-11-23
**General Response to All the Reviewers**

We sincerely thank all the reviewers for their insightful and constructive feedback on our manuscript. We are encouraged to see positive recognition of our work:

- **Writing is generally clear / well organized and clearly written.**  ($\frac{2}{4}$ reviewers: J6jt, jCwd)
- **Comprehensive and systematic evaluation / Broad coverage.** ($\frac{4}{4}$  reviewers: 5GKb,  J6jt, jCwd, yQT6)
- **Significant benchmarking contribution / Practical contribution.** ($\frac{3}{4}$ reviewers: 5GKb, J6jt, jCwd)

We have carefully addressed all the reviewers' comments and provided detailed responses to each point. The revised manuscript has been submitted to the system. If there are any further questions or concerns, please don’t hesitate to reach out. We remain committed to improving the quality of this work and welcome further discussions.

Thank you once again for your valuable feedback and support!

---

### Note · Program_Chairs · 2026-01-17
**Submission Desk Rejected by Program Chairs**

The following references in this submission do not refer to real documents and/or have major errors in bibliographic information:

 Zhipeng et al. He. An atlas of human brain organoids. Cell, 187(22):5701-5721.e12, 2024. doi: 10.1016/j.cell.2024.09.036.
Rui Yang and Paula Gomez. cell-o1: Batch-level reasoning for single-cell rna-seq cell type annotation. bioRxiv, 2024.